



# An integrated analysis of contemporary methane emissions and concentration trends over China using in situ, satellite observations, and model simulations

Haiyue Tan[1], Lin Zhang[1], Xiao Lu[2], Yuanhong Zhao[3], Bo Yao[4], Robert J. Parker[5,6], Hartmut Boech[5,6]

[1]Department of Atmospheric and Oceanic Sciences, School of Physics, Peking University, Beijing, China
[2]School of Atmospheric Sciences, Sun Yat-sen University, Zhuhai, China
[3]College of Oceanic and Atmospheric Sciences, Ocean University of China, Qingdao, China
[4]Meteorological Observation Centre of China Meteorological Administration (MOC/CMA), Beijing, China
[5]National Centre for Earth Observation, University of Leicester, Leicester, UK
[6]Earth Observation Science, School of Physics and Astronomy, University of Leicester, UK

*Correspondence to*: Lin Zhang (zhanglg@pku.edu.cn)

**Abstract.**

China, being one of the major emitters of greenhouse gases, has taken strong actions to tackle climate change, e.g., to achieve carbon neutrality by 2060. It also becomes important to better understand the changes in the atmospheric mixing

ratio and emissions of $CH_4$, the second most important human-influenced greenhouse gas, in China. Here we analyze the sources contributing to the atmospheric $CH_4$ mixing ratio and their trends in China over 2007–2018 using the GEOS-Chem model simulations driven by two commonly used global anthropogenic emission inventories: the Emissions Database for Global Atmospheric Research (EDGAR v4.3.2) and the Community Emissions Data System (CEDS). The model results are interpreted with an ensemble of surface, aircraft, and satellite observations of $CH_4$ mixing ratios over China and the Pacific

region. The EDGAR and CEDS estimates show considerable differences reflecting large uncertainties in estimates of Chinese $CH_4$ emissions. Chinese $CH_4$ emission estimates based on EDGAR and natural sources increase from 46.7 Tg per annum (Tg $a^{-1}$) in 1980 to 69.8 Tg $a^{-1}$ in 2012 with an increase rate of 0.7 Tg $a^{-2}$, and estimates with CEDS increase from 32.9 Tg $a^{-1}$ in 1980 and 76.7 Tg $a^{-1}$ in 2014 (a much stronger trend of 1.3 Tg $a^{-2}$ over the period). Both surface, aircraft, and satellite measurements indicate $CH_4$ increase rates of 7.0–8.4 ppbv $a^{-1}$ over China in the recent decade. We find that the

model simulation using the CEDS inventory and interannually varying OH levels can best reproduce these observed $CH_4$ mixing ratios and trends over China. Model results over China are sensitive to the global OH level, with a 10% increase in the global tropospheric volume-weighted mean OH concentration presenting a similar effect to that of a 47 Tg $a^{-1}$ decrease in global $CH_4$ emissions. We further apply a tagged tracer simulation to quantify the source contributions from different emission sectors and regions. We find that domestic $CH_4$ emissions account for 11.4% of the mean surface mixing ratio and

drive 68.3% of the surface trend (mainly via the energy sector) in China over 2007–2018. We emphasize that intensive $CH_4$ measurements covering eastern China will help better assess the driving factors of $CH_4$ mixing ratios and support the emission mitigation in China.



# 1 Introduction

Atmospheric methane ($CH_4$) is the second most important anthropogenic greenhouse gas contributing more than a quarter of the human-induced radiative imbalance since 1750 (IPCC, 2013). It also plays an important role in atmospheric chemistry as an essential precursor for tropospheric ozone and stratospheric water vapor (Turner et al., 2019). Global mean atmospheric $CH_4$ surface concentrations increased from about 1650 ppbv in the mid 1980s to about 1770 ppbv in the late 1990s, then stabilized around this level in the early 2000s, and started increasing again since 2007 (Dlugokencky et al., 2009; Nisbet et

al., 2019). The regrowth of atmospheric $CH_4$ concentrations has drawn worldwide attention and led to many different or even contradictory explanations (Maasakkers et al., 2019; Turner et al., 2019; Zhao et al., 2019; Yin et al., 2020; Zhang et al., 2021). Difficulties in the attribution of the trends are mainly associated with large uncertainties in changes in the $CH_4$ emissions from various sources as well as the chemical loss via oxidation by hydroxyl radical (OH) (Turner et al., 2019). A better understanding and quantification of the interannual variability of $CH_4$ emissions and the drivers of the concentration

growth in the recent decade is important to support its mitigation.

$CH_4$ has both important anthropogenic and natural sources. It can be emitted from human activities including coal mining, oil and gas exploitation, livestock, rice cultivation, waste deposit, and wastewater treatment. It also has a large natural source from wetlands, with small sources from forest fires, termites, and geological seeps. Global bottom-up estimates of $CH_4$

emissions based on statistics of source activities or process-based models have reported a wide range of total $CH_4$ emissions of 542–852 Tg $a^{-1}$ in the 2000s (Kirschke et al., 2013). Atmospheric top-down analyses constrained by surface, satellite, and aircraft observations of $CH_4$ concentrations tend to suggest lower total $CH_4$ emissions of 526–569 Tg $a^{-1}$ in the period (Kirschke et al., 2013) and find even greater uncertainties in the relative contributions from different $CH_4$ emission sectors (Kirschke et al., 2013; Saunois et al., 2016; Saunois et al., 2020). Over 90% of atmospheric $CH_4$ is lost via oxidation by OH

in the troposphere, leading to a lifetime of 9.14 (±10%) years against this sink. Additional minor sinks include soil absorption, loss in the stratosphere, and reactions with chlorine radicals (IPCC, 2013). The contemporary growth of atmospheric $CH_4$ levels reflect an imbalance between its global sources and sinks.

China is one of the most significant methane producers, especially for anthropogenic sources such as coal mining (Saunois et

al., 2016). Anthropogenic sources in China contribute about 13% of the global anthropogenic $CH_4$ emissions in the 2000s (Kirschke et al., 2013). The recent bottom-up emission inventory of Peng et al. (2016) found that the total Chinese $CH_4$ emissions increased from 24.4 Tg $a^{-1}$ in 1980 to 45.0 Tg $a^{-1}$ in 2010, with the largest source sector being rice cultivation in 1980 and replaced by coal mining after 2005. However, large uncertainties exist in our understanding of the contemporary changes of $CH_4$ emissions over China (Saunois et al., 2020), e.g., whether the Chinese $CH_4$ emissions from coal mining has



decreased due to the mitigation policy in recent years (Miller et al., 2019; Sheng et al., 2019). Atmospheric inversion
      analyses are typically applied at global scales due to very limited in situ $CH_4$ measurements over this region in 2000s. The
      increases of spatiotemporal observations (from satellite or aircraft) and the development of atmospheric transport models
      would be helpful in constraining methane sources over China, but different dataset and methods could provide discrepant
      information (Thompson et al., 2015; Miller et al., 2019). China has pledged to peak the carbon dioxide emissions by 2030
and to reach carbon neutrality by 2060 for tackling climate change. As $CH_4$ being the second most important anthropogenic
      greenhouse gas, it also becomes crucial to quantify its emissions and concentration trends in China.

      In this study, we aim to better understand the recent trends in $CH_4$ emissions and concentrations in China using the GEOS-
      Chem (Goddard Earth Observing System-Chemistry) chemical transport model driven by two commonly used global
anthropogenic emission inventories: the Emission Database for Global Atmospheric Research (EDGAR, version 4.3.2)
      (Janssens-Maenhout et al., 2019) and the Community Emissions Data System (CEDS, version 2017-05-18) (Hoesly et al.,
      2018). We use an ensemble of surface, aircraft, and satellite observations to assess the $CH_4$ concentrations and trends from
      surface to the troposphere, and conduct a series of model simulations to examine their driving factors as well as the influence
      of the interannual variability of global volume-weighted OH concentrations. An improved tagged $CH_4$ tracer simulation
(with 100 region- and sector-specific tracers) is applied to identify and quantify the contributions to the spatial patterns of
      $CH_4$ concentrations and trends over China in the recent decade of 2007–2018.

## 2 Measurements and the GEOS-Chem model

### 2.1 Surface and aircraft measurements

      We use the surface $CH_4$ concentration measurements from the Global Monitoring Division (GMD) of the Earth System
Research Laboratory (ESRL) at the National Oceanic and Atmospheric Administration (NOAA). The $CH_4$ concentrations are
      measured by gas chromatography with flame ionization detection (Dlugokencky, 2005). The measurement database
      (https://www.esrl.noaa.gov/gmd/dv/data/, last access: 3 March 2021) includes 95 sites globally providing monthly averages
      of mixing ratios (ppbv). The database has been widely used in assessing regional and global $CH_4$ concentrations and budgets
      (Bergamaschi et al., 2013; Fraser et al., 2013; Cressot et al., 2014; Turner et al., 2016; Miller et al., 2019).


      Here we focus on four sites located in China, as summarized in Table 1, including Dongsha Island (DSI, 20.7$^{\circ}$ N, 116.7$^{\circ}$ E)
      measuring from March 2010 to December 2018, Lulin (LLN, 23.5$^{\circ}$ N, 120.9$^{\circ}$ E) from August 2006 to December 2018,
      Shangdianzi (SDZ, 40.7$^{\circ}$ N, 117.1$^{\circ}$ E) from September 2009 to September 2015, and Waliguan (WLG, 36.3$^{\circ}$ N, 100.9$^{\circ}$ E)
      from May 1991 to December 2018. Three of these sites (LLN, SDZ, and WLG) are mountain-top sites while DSI is located


in the marine boundary layer. The WLG site located in the Qinghai-Tibet Plateau at 3810 m above sea level is the first
baseline observatory in China, providing continuous measurements since the year 1991.

We analyze measurements of $CH_4$ concentrations from two aircraft campaigns: the High-performance Instrumented Airborne
Platform for Environmental Research (HIAPER) Pole-to-Pole observation (HIPPO) and the Atmospheric Tomography
Mission (ATom). HIPPO consists of five campaigns from January 2009 to September 2011 (Wofsy et al., 2011). ATom
consists of four campaigns from July 2016 to May 2018 (Wofsy et al., 2018). Figure 1 shows the flight tracks from two
campaigns. Both HIPPO and ATom datasets provide the merged 10-second data products for all flights (Wofsy et al., 2017;
Wofsy et al., 2018), which cover the four seasons temporally and the regions over the Pacific Ocean and North America
spatially. Both campaigns conduct continuous profiling between ~0.15 km and 8.5 km altitude, with many profiles extending
to nearly 14 km. Here we sample the model results at the hourly resolution along flight tracks as shown in Fig. 1 and average
them in 2° latitude bins for the comparison.

## 2.2 GOSAT satellite observations

The TANSO-FTS instrument onboard the Greenhouse Gases Observing Satellite (GOSAT) launched in early 2009 measures
the backscattered solar radiation from a sun-synchronous orbit at around 13:00 local time (Butz et al., 2011; Kuze et al.,
2016). The observations have a pixel resolution of around 10 km diameter and are separated by about 250 km along the
observing track with a global coverage every 3 days (Parker et al., 2015). GOSAT retrieves column-averaged dry-air $CO_2$
and $CH_4$ mixing ratios from the shortwave infrared (SWIR) spectrum with near-unit sensitivity down to the surface (Butz et
al., 2011). We use the University of Leicester version 7.2 GOSAT $XCH_4$ proxy retrieval over China from January 2010 to
December 2017. The glint data over the oceans are not used in this study due to the sparse data coverage. The $CH_4$ product
has been validated by Parker et al. (2015) against the Total Carbon Column Observing Network (TCCON) and MACC-II
model XCH4 data and suggested a precision of 0.7%.

To compare with the GEOS-Chem model results as described below, the GOSAT $CH_4$ observations and satellite averaging
kernels are averaged over the 2°×2.5° or 4°×5° model grid. We use the satellite observations which pass the criteria that the
grid has more than 12 months of valid observations which have passed their quality control. The simulated vertical profiles
($VMR^{mod}$) are applied with the satellite averaging kernels (AK) and a priori estimates ($VMR^{apr}$) using Equ. (1) following
Parker et al. (2020).

$$XCH_4^{mod} = \sum_{i=0}^{N_{lev}}\{[VMR_i^{apr} + (VMR_i^{mod} - VMR_i^{apr})AK_i]h_i\} \quad (1)$$

where $AK_i$ is the retrieval averaging kernel and $h_i$ is the pressure weight for the vertical level $i$. This provides column mean
$CH_4$ mixing ratios ($XCH_4^{mod}$) with the vertical sensitivity of satellite retrievals accounted for.



## 2.3 The GEOS-Chem model description and simulation design

We use the GEOS-Chem global chemical transport model v11-02 release candidate (http://geos-chem.org, last access: 3 March 2021) driven by MERRA-2 meteorological fields from the NASA Global Modeling and Assimilation Office
(GMAO). The MERRA-2 dataset has a native horizontal resolution of 0.5° latitude×0.625° longitude, and is degraded to 4°×5° or 2°×2.5° resolutions for input to GEOS-Chem. We use the $CH_4$ simulation that calculates the $CH_4$ sinks using prescribed global distributions of OH concentrations or loss frequencies. The model has been applied in a number of studies to understand the global and regional $CH_4$ emissions and concentrations (Wecht et al., 2014; Turner et al., 2015; Maasakkers et al., 2019; Lu et al., 2021; Maasakkers et al., 2021; Zhang et al., 2021).


We use and compare two global anthropogenic $CH_4$ inventories: the Emissions Database for Global Atmospheric Research (EDGAR v4.3.2) covering 1970–2012 (Janssens-Maenhout et al., 2019) and the Community Emissions Data System (CEDS, version 2017-05-18) (Hoesly et al., 2018) covering 1970–2014. A detailed comparison of the two emission estimates will be presented in Section 3. The EDGAR $CH_4$ emissions do not account for seasonal variations. Here we have applied seasonal
scalars to $CH_4$ emissions from manure management based on a temperature dependence described by Maasakkers et al. (2016) and to those from rice cultivation following Zhang et al. (2016) in the EDGAR inventory. The CEDS inventory as used in this study provides gridded emission estimates with monthly variations.

For natural sources, monthly wetland emissions are from the WetCHARTs version 1.0 extended ensemble mean for 2001–
2015 (Bloom et al., 2017) and are scaled by 1.1 to match the estimates in Kirschke et al. (2013) and Saunois et al. (2020). Open fire emissions are from the Quick Fire Emissions Database version 2.4 with daily variability over 2009–2015 (Darmenov and da Silva, 2013). Termite and seepage emissions are, respectively, from Fung et al. (1991) and Maasakkers et al. (2019).

The oxidation of $CH_4$ by tropospheric OH is calculated in the model using 3-D monthly averaged OH concentrations archived from a standard GEOS-Chem tropospheric chemistry simulation in Wecht et al. (2014). Global uniform scalers are then applied to account for the interannual variability of OH concentrations during 1980–2010 as simulated by the CESM model in Zhao et al. (2019). As shown in Fig. S1, the resulting global volume-weighted mean OH increases by 0.20% $a^{-1}$ in 1980–2000 and 0.37% $a^{-1}$ in 2000–2010, finally reaching to $10.9\times10^5$ molecules $cm^{-3}$. Other minor sinks include
tropospheric oxidation by chlorine atoms using monthly chlorine concentration fields of Sherwen et al. (2016), stratospheric loss computed with monthly loss frequencies of Murray et al. (2012), and soil uptake of Fung et al. (1991) with a temperature dependent seasonality (Ridgwell et al., 1999).



We have conducted a series of model simulations over 1980–2018 as summarized in Table 1 to investigate the impacts of OH concentrations and model resolution. For all the datasets of emissions and sinks as described above, the closest available year will be used for simulation years beyond their available time ranges. Evaluations of these model results with the NOAA surface measurements at the four Chinese sites indicate that the simulation with CEDS and interannually varying OH at 2°×2.5° resolution (GCC in Table 1) relatively better captures the measured concentrations and trends since 2007, as will be discussed in Section 3.2.

We further apply a tagged $CH_4$ tracer simulation to quantify the sources contributing to $CH_4$ concentrations and trends in China over 2007–2018. We implement 100 tracers that tag $CH_4$ emissions from different source types (agriculture, energy, industry, transportation, wastewater, residents, shipping, biomass burning, wetlands, seeps and termites) and different regions (China, India, Europe, South America, North America, Africa, Oceania, etc., as shown in Fig. 2). Global soil uptake is also tagged as a sink of $CH_4$. We run the tagged $CH_4$ simulation using the model settings of GCC (i.e., CEDS and interannually varying OH) for the period of 1980–2018. The results allow us to quantify the detailed source contributions to $CH_4$ concentrations and trends over China.

## 3 Results

### 3.1 $CH_4$ emissions and sinks over the globe and China

Figure 3 and Supplementary Table S1 compare the anthropogenic emissions of EDGAR and CEDS, natural emissions, and sinks in our model simulations (GCE and GCC in Table S1) with the estimates in the literature summarized by Saunois et al. (2020). The emissions in the two decades of 2000–2009 and 2008–2017 from both bottom-up and top-down studies are reported in Saunois et al. (2016; 2020), and are thus compared with corresponding estimates in this study. The anthropogenic emission source categories are different in the EDGAR and CEDS inventories, and we organize all sources into five main categories (agriculture and waste, biomass burning, fossil fuels, wetlands, and other sources) following Saunois et al. (2020), as also summarized in Table S2.

As shown in Fig. 3, the global total (anthropogenic and natural) emissions over 2000–2009 are 520 Tg a$^{-1}$ for GCE and 533 Tg a$^{-1}$ for GCC. These total emissions are in the low end of the top-down estimates of 547 Tg a$^{-1}$ with a range of 524–560 Tg a$^{-1}$, and are smaller than the bottom-up estimates of 703 (566–842) Tg a$^{-1}$. The bottom-up estimates summarized by Saunois et al. (2020) included EDGAR and CEDS, and we can see that the differences with our emissions are largely driven by the underestimates of some natural emissions (e.g., geological, termites, and freshwaters emissions), which are substantially reduced in the top-down estimates. In the 2008–2017 period, global total $CH_4$ emissions in GCE and GCC have increased to 556 Tg a$^{-1}$ in GCE and to 574 Tg a$^{-1}$ in GCC, and are within the top-down emission range of 576 (550-594) Tg





$a^{-1}$. The contributions of anthropogenic sources on total $CH_4$ emissions are about 63% (2000–2009) and 65% (2008–2017) in GCE, and 65% (2000–2009) and 67% (2008–2017) in GCC, which are slightly larger than 60% and 62% in the top-down estimates of Saunois et al. (2020). The global $CH_4$ chemical losses simulated in GCE and GCC are also consistent with the top-down estimates for both periods, while the sink of soil uptake might be underestimated in the model.

Table 2 and Fig. 4 compare the annual $CH_4$ emissions and sinks in China simulated in GCE and GCC with the results reviewed by Saunois et al. (2020) and Kirschke et al. (2013) and a bottom-up anthropogenic emission inventory of Peng et al. (2016) for the period of 2000–2009. Total Chinese $CH_4$ emissions are 57.2 Tg $a^{-1}$ (2000–2009) and 67.6 Tg $a^{-1}$ (2008–2017) in GCE, and 55.5 Tg $a^{-1}$ (2000–2009) and 73.7 Tg $a^{-1}$ (2008–2017) in GCC. Considerable differences between GCE and GCC can be seen for the emission estimates of different sectors. The $CH_4$ emissions from fossil fuels over 2000–2009 are

23.4 Tg $a^{-1}$ in GCC, which are at the high end of the bottom-up estimates (12.6–23.9 Tg $a^{-1}$) summarized in Saunois et al. (2020). The $CH_4$ emissions from fossil fuels in GCE are smaller (15.8 Tg $a^{-1}$ over 2000–2009), and are slightly higher than the estimate of 12.8 in Peng et al. (2016). By contrast, $CH_4$ emissions from agricultural and waste in GCE (33.3 Tg $a^{-1}$ over 2000–2009) are much higher than those in GCC (25.3 Tg $a^{-1}$ over 2000–2009), and they are, respectively, at the high and low ends of the bottom-up (24.0–33.0 Tg $a^{-1}$) estimates in Saunois et al. (2020). The natural sources (e.g., wetlands, biomass

burning) and the soil uptake in our study are relatively low compared with the estimates in Saunois et al. (2020). For the period of 2008–2017, the $CH_4$ emissions from fossil fuels increase to 22.8 Tg $a^{-1}$ in GCE and 38.4 Tg $a^{-1}$ in GCC, which are also at the averaged level and the high end of the bottom-up estimate (16.6–39.6 Tg $a^{-1}$) in Saunois et al. (2020).

Figure 5 further shows annual total Chinese $CH_4$ emissions from different sectors and their percentage contributions during

1980–2018 in both GCE and GCC simulations. Chinese total $CH_4$ emissions in GCE are 46.7 Tg $a^{-1}$ in the year 1980 and increase to 69.8 Tg $a^{-1}$ (49.5% increase) in 2012 (the last available year for EDGAR v4.3.2), presenting an increase trend of 0.7 Tg $a^{-2}$ over 1980–2018. GCC simulations have a stronger trend of 1.3 Tg $a^{-2}$ over the period than GCE, with total emissions of 32.9 Tg $a^{-1}$ in 1980 and 76.7 Tg $a^{-1}$ in 2014 (the last available year of CEDS). Both GCE and GCC show faster increases after 2003 than the years before, which are largely driven by the emissions from the fossil fuels or energy sector.

The largest differences between GCE and GCC, as also discussed in Fig. 4, come from the sectors of fossil fuels and agriculture. Agriculture sources in GCE account for 54.7% of the total $CH_4$ emissions in 1980 and gradually decrease to 37.0% in 2018, which mainly result from decreases in emissions from rice cultivation with some offset due to increases in the livestock emission. The contributions of agricultural sources in GCC are much smaller with values of 36.3% in 1980 and 21.5% in 2018. The energy or fossil fuels sector becomes the largest contributor of Chinese $CH_4$ emissions in recent years in

GCC, accounting for 52.2% of the total emissions in 2018, and largely drives the larger positive trend in GCC than GCE.





The comparisons above indicate large uncertainties in the Chinese $CH_4$ emission estimates, as to some extent covered by the EDGAR and CEDS anthropogenic emission inventories. The magnitude and temporal variations of methane budgets over the past decades are known to have large uncertainties (Kirschke et al., 2013; Turner et al., 2019; Saunois et al., 2020).

Relative uncertainties are about 20–35% for anthropogenic emissions such as fuel exploitation, agriculture and waste, about 50% for biomass burning and wetlands, and reach 100% or greater for other natural sources (Saunois et al., 2020). Uncertainties in the methane sinks are about 10–20% by proxy methods such as using methyl chloroform, and are 20–40% by atmospheric chemistry models (Saunois et al., 2016). More detailed regional methane datasets can help improve assessing the global budget (Xu and Tian, 2012; Valentini et al., 2014; Saunois et al., 2016). We will further discuss the uncertainties

in $CH_4$ emissions in the last section.

## 3.2 Observed and simulated methane concentrations and trends in China

Based on the emissions described above, we have conducted a series of model simulations as summarized in Table 1 and evaluated the model results with surface $CH_4$ measurements at the four Chinese sites. We find that when using the interannually fixed OH (global tropospheric volume-weighted mean of $10.6 \times 10^5$ molecules $cm^{-3}$ as shown in Fig. S1), both

model simulations with the EDGAR and CEDS emissions overestimate the observed $CH_4$ trends since 2007 by 0.8–6.2 ppbv $a^{-1}$ with EDGAR (Run1) and by 4.0–10.9 ppbv $a^{-1}$ with CEDS (Run2). The model simulated $CH_4$ concentrations and trends over China are rather sensitive to the global OH levels. In the sensitivity simulations with global OH decreasing 10% (Run5) or increasing 10% (Run6) relative to the fixed levels (global mean of $10.9 \times 10^5$ molecules $cm^{-3}$) over 2010–2018, $CH_4$ concentrations would, respectively, increase by 2.0%–3.4% or decrease 1.9%–3.2% at the four Chinese sites (Fig. S2).

Increasing OH levels by 10% would lead negative trends in $CH_4$ concentrations at all four sites over 2010–2018 (Fig. S2). Such effects are also found in the simulation with global $CH_4$ emissions decreasing 50 Tg $a^{-1}$ over the same period (Run 7 in Table 1 and Fig. S2).

The uses of interannually varying OH (Fig. S1) in model simulations (Run3 and Run4 in Table 1) overall correct the high

biases in simulated $CH_4$ trends in simulations with fixed OH (Run1 and Run2) at the Chinese sites. We find that changing model horizontal resolution from 4°×5° to 2°×2.5° does not significantly affect the simulated surface $CH_4$ trends. Hereafter, we will focus our analyses on the model simulations at 2°×2.5° resolution and with interannually varying OH (i.e., GCE and GCC in Table 1).

Figure 6 shows the measured and simulated time series of monthly $CH_4$ concentrations at the four Chinese sites. Both GCE and GCC model results are shown, and distinct differences in $CH_4$ concentrations can be seen between the two simulations. Among the four Chinese sites, the largest $CH_4$ concentrations are observed at the SDZ site, a rural site near Beijing surrounded by high anthropogenic emissions, compared with the other three Chinese background sites (DSI, LLN, and



WLG). GCC with high anthropogenic emission estimates simulate on average 1.0%–4.7% higher $CH_4$ concentrations than
GCE results, and are 0.3%–6.5% higher than measurements at the four Chinese sites. Measured $CH_4$ concentrations at the
four sites are increasing at the rates of 7.0–7.9 ppbv $a^{-1}$ in recent years since 2007. The GCC model results reproduce the
trends in $CH_4$ concentrations at the DSI, LLN, and WLG sites, while overestimate the 2009–2015 trend measured at SDZ by
a factor of two. The GCE model results in general underestimate the measured trends except for that at the SDZ site. These
results can be explained by the higher $CH_4$ emission estimates and trends in CEDS than EDGAR since 2007, and may also
reflect the regional $CH_4$ emissions around SDZ are too high in CEDS.

Comparisons with satellite and aircraft observations further provide spatially and vertically resolved evaluations of the
model simulations. Figure 7 and 8 show, respectively, the GOSAT observed and model simulated spatial distributions of
seasonal mean $CH_4$ concentrations and trends over 2010–2017. The latitude-dependent biases between simulations and
observations have found noticeable at the 4°×5° resolution, but is significantly smaller at 2°×2.5° (Stanevich et al., 2020).
The GOSAT observed $CH_4$ column concentrations over China peak in autumn (1825.6 ppbv on average) and reach a
minimum in spring (1797.4 ppbv). There is a stronger seasonality in the $CH_4$ concentration in the South China (1856.9 ppbv
in autumn vs. 1826.8 ppbv in spring) likely attributed to the seasonal variation in agriculture emissions. The GOSAT
observed 2010–2017 trends show small spatial and seasonal variations over China with values of 7.67–8.43 ppbv $a^{-1}$. Both
GCE and GCC model simulated $CH_4$ concentrations present similar spatial patterns with high correlation coefficients (r >
0.90), while GCE simulated concentrations are on average biased low by 23.5–32.4 ppbv (~1.6%), and GCC results are
overestimated by 25.6–36.8 ppbv (~1.7%). This discrepancy between the two simulations is mainly due to the $CH_4$ emissions
from fossil fuels, which are 23.5 Tg $a^{-1}$ for the GCE and 39.9 Tg $a^{-1}$ for the GCC in China over 2010–2017. GCC model
results better capture the observed 2010–2017 $CH_4$ trends over China with small biases of −1.7–0.4 ppbv $a^{-1}$, compared to
the GCE results that in general underestimate the trends by 2.6–4.7 ppbv $a^{-1}$.

Figure 9 shows the latitudinal distribution of annual mean $CH_4$ concentrations as observed by HIPPO and ATom aircraft
campaigns at three altitude layers (1–2 km, 4–5 km, and 7–8 km). Model results sampled along the flight tracks at their
observing time are also shown. Both aircraft measurements and model results are then averaged in 2º latitude bins. As shown
in Fig. 9, large latitudinal gradients in the tropospheric $CH_4$ concentration between the northern and southern hemispheres, in
particular in the lowest 2 km of the tropics, are observed by the aircraft measurements, and are captured by the model results
with the two emission inventories. Both GCE and GCC model simulated $CH_4$ concentrations present consistent with the
GOSAT biases due to the lower estimate of global emissions in GCE (556 Tg $a^{-1}$) than GCC (574 Tg $a^{-1}$) since 2008 as
shown in Table S1. GCE model results underestimate the aircraft measurements with mean negative biases of 27.5–31.1
ppbv at the three altitude layers for HIPPO, and even larger negative biases of 61.5–73.7 ppbv for ATom. By contrast, GCC



model results are in general too high with biases of 18.4–22.8 ppbv for HIPPO, and −1.7–9.4 ppbv for ATom. The biases in GCC are overall smaller than those in GCE.

The changes in the model bias for the comparisons with HIPPO and ATom measurements reflect their simulated trends in the $CH_4$ concentration. Since both HIPPO (2009–2011) and ATom (2016–2018) provide measurements over the Pacific (black box in Fig. 1), we calculate the differences between HIPPO and ATom measurements as the observed $CH_4$ concentration trends over this region. Figure 10 shows aircraft observed and corresponding model simulated trends separated for four seasons. The HIPPO (2009–2011)–ATom (2016–2018) $CH_4$ trends as estimated by the aircraft measurements range 5.8–10.7 ppbv a$^{-1}$ for the different seasons and altitudes, with typically higher increasing rates in boreal summer and autumn
than those in boreal spring and winter. Both GCE and GCC model results tend to underestimate the trends, but the biases in GCC are much smaller than GCE. A distinct feature can be seen from aircraft observations is the high $CH_4$ increasing rates over the tropics in boreal summer and autumn (reaching 15 ppbv a$^{-1}$), while both model results do not capture it and show weak latitudinal gradients in the $CH_4$ trends. These tropical $CH_4$ increases are likely driven by the increasing tropical microbial emissions either from wetlands or livestock shown in some recent papers (Nisbet et al., 2016; Saunois et al., 2017;
Worden et al., 2017; Maasakkers et al., 2019; Yin et al., 2020; Zhang et al., 2021), which have not been found in the model simulations.

Summarizing the comparisons of model results with all available measurements over China and the Pacific, we find that the surface, aircraft, and satellite $CH_4$ measurements have indicated rather consistent increase rates of $CH_4$ concentrations over
China with values ranging 7.0–8.4 ppbv a$^{-1}$ in recent years. As $CH_4$ has a lifetime of about 9.14 years, such increases reflect changes in not only domestic emissions but also global emissions. The GCE and GCC model simulations with the interannually varying OH levels both capture the main features of the observed $CH_4$ concentrations and trends over China, and the GCC results show much smaller model biases than GCE. We will thus use the GCC model simulation to quantify the domestic and global sources contributing to the $CH_4$ concentrations and trends over China.

**3.3 Source attribution of $CH_4$ concentrations and trends in 2007–2018**

Here we apply the GCC model configuration (i.e., the CEDS inventory and interannually varying OH) in the tagged $CH_4$ simulation. The GCC model results can generally reproduce the spatial distribution of GOSAT observed $CH_4$ levels and trends as shown in Fig. S3, with mean biases of 27.4 ppbv (observed 1805 ppbv vs. simulated 1833 ppbv) in the global $CH_4$ concentration and −0.8 ppbv a$^{-1}$ (observed 7.08 ppbv a$^{-1}$ vs. simulated 6.26 ppbv a$^{-1}$) in the trend. As described in the
Section 2.3, our tagged $CH_4$ simulation includes 100 region- and sector-specific $CH_4$ tracers. The tagged $CH_4$ simulation is conducted over 1980–2018, and we analyze the results for 2007–2018.  Figure 11 shows contributions of $CH_4$ emissions from different source regions and different sectors on the mean surface concentrations and trends in China during this time





period, and the values are also summarized in Table 3 for concentrations and Table 4 for trends. As for the concentrations, we find that the largest contributor of Chinese $CH_4$ concentrations averaged over 2007–2018 is the wetland emission in

South America, accounting for 11.2% due to the large emission magnitude. Together with other sources, emissions in South America contribute 20.6% of the surface $CH_4$ levels over China, followed by the sources from Europe (17.0%), Africa (16.6%), North America (13.9%), and Rest Asia (12.6%). The Chinese domestic emissions account for 11.4% of the $CH_4$ concentrations. The emission contributions to the concentrations are generally proportional to their emission magnitudes because of the $CH_4$ lifetime of about 9.14 years, and seasonal variations in the percentage contributions are small as can be

seen in Fig. 11 (the top left panel).

Figure 11 and Table 4 also show the source contributions to the 2007–2018 trends in surface $CH_4$ concentrations over China. Based on the emission inventory in GCC, the simulated mean trend in the surface $CH_4$ concentration is 8.32 ppbv $a^{-1}$ over the land of China. The domestic energy sector is identified as the largest driver of the trend in China contributing an increase

rate of 4.32 ppbv $a^{-1}$. Accounting for the trends driven by emissions from agriculture and wastewater sectors, domestic contributions can reach 5.68 ppbv $a^{-1}$ (68% of 8.32 ppbv $a^{-1}$). The remaining trends of 2.64 ppbv $a^{-1}$ are then contributed by emission changes outside China. We find that the anthropogenic sources (mainly from energy, agriculture and wastewater sectors) in Africa and other Asian regions (India and Rest Asia) contribute, respectively, trends of 3.29 and 2.40 ppbv $a^{-1}$ over China, highlighting the strong $CH_4$ emission increases in these regions such as large emission increases mainly from

livestock sources over South Asia and tropical Africa in 2010–2018 (Zhang et al., 2021). On the contrary, Europe is the only region where $CH_4$ emissions from nearly all sectors have been decreasing (Jackson et al., 2020), which lead to a negative trend of −3.56 ppbv $a^{-1}$ over China. Not only near the surface, we find similar results for the $CH_4$ concentrations throughout the troposphere over China with slightly smaller growth rates in the upper troposphere (Fig. S4).

Our results indicate that trends in China are dominated by energy emissions from coal, oil and gas, with significant contributions from wastewater and agriculture sectors. This is consistent with the top-down emission inversion results by Miller et al., (2019) that found the Chinese coal emission is increasing in 2010–2015, while the bottom-up emission estimates of Sheng et al. (2019) suggested decreases in the coal emission in 2012–2016. The lack of sub-country emission factors may result in large uncertainties in the bottom-up emission estimates. A recent global emission inversion study using

the EDGAR v4.3.2 inventory as the prior estimate also found large overestimates in the Chinese emissions from coal (Maasakkers et al., 2019). Using the overestimated emissions from the domestic coal sector in the model would offset the influence of missing increases in microbial emissions in the tropics as discussed in Section 3.2.

The analyses above demonstrate strong foreign source contributions to the $CH_4$ concentrations as well as $CH_4$ trends over

China. We further find large spatial heterogeneity in the domestic vs. foreign contributions. Figure 12 shows the spatial





distributions of domestic emission contributions to Chinese $CH_4$ surface concentrations and trends over 2007–2018 calculated as the percentages of sums of all Chinese tagged tracers to the total levels. We can see that the domestic contributions to the $CH_4$ surface concentration range from 10.2% in the western China to 13.0% in central China, and to the trends range from 54.6% over the Tibet Plateau to 74.4% in the central China. The largest domestic contributions for both

surface concentrations and trends are found in the central eastern China, so that measurements over this region would most reflect the $CH_4$ emission changes in China.

## 4 Conclusions and discussion

In summary, we have investigated the sources contributing to the $CH_4$ concentrations and trends over China in the recent decade (2007–2018) using the GEOS-Chem global model. The $CH_4$ model simulations are conducted considering two

different commonly used anthropogenic emission inventories (EDGAR v4.3.2 and CEDS), and are evaluated with available surface, aircraft, and satellite measurements of $CH_4$ concentrations over China and the Pacific region. The surface, aircraft, and satellite measurements have shown $CH_4$ concentration increase rates of 7.0–8.4 ppbv $a^{-1}$ over China in recent years. We find that model results are sensitive to the selection of anthropogenic emission inventories and OH levels. By using the CEDS anthropogenic emission inventory and interannually varying OH levels (Fig. S2) the model can generally reproduce

the measured $CH_4$ concentrations and trends over China. This corresponds to mean Chinese anthropogenic $CH_4$ emissions of 69.4 Tg $a^{-1}$ (with an increase rate of 1.2 Tg $a^{-2}$), and global tropospheric volume-weighted mean OH concentrations of $10.8 \times 10^5$ molecule $cm^{-3}$ (with an increase rate of 0.25% $a^{-1}$) over 2007–2018.

We apply a tagged $CH_4$ tracer simulation that implements region- and sector-specific tracers to diagnose and to understand

their emission contributions. Using the model simulation with CEDS and interannually varying OH, we find strong influences from foreign sources on both $CH_4$ concentrations and recent increases over China as $CH_4$ as a relatively long lifetime of about 9.14 years. For the mean surface $CH_4$ concentration over China (1873.0 ppbv over 2007–2018), domestic $CH_4$ emissions account for 11.4%, and contributions from the sources outside China reaching 88.6%, including 20.6% from South America, 17.0% from Europe, 16.6% from Africa, 13.9% from North America, and 12.6% from the Rest Asia. For the

mean $CH_4$ concentration trend over China (8.32 ppbv $a^{-1}$ over 2007–2018), the largest driver is estimated to be the domestic energy source contributing 4.32 ppbv $a^{-1}$, and other important domestic source contributions include emissions from wastewater (1.00 ppbv $a^{-1}$) and agriculture (0.53 ppbv $a^{-1}$); natural sources such as wetland emissions have insignificant trend contributions. Emission changes in foreign sources are also significant. The increase rate of 3.20 ppbv $a^{-1}$ in the Chinese surface $CH_4$ concentration can be attribute to sources in Africa, 2.20 ppbv $a^{-1}$ to other Asian countries (India and

Rest Asia), and 1.39 ppbv $a^{-1}$ to South America (Table 4).



It shall be noted that our source attribution results can be biased by the use of CEDS and the uncertainty in the interannual variations of OH levels. The Chinese anthropogenic $CH_4$ emissions in the CEDS inventory are higher and increase more rapidly than EDGAR v4.3.2 in the recent decade. The two emission inventories significantly differ in the sectors of fossil

fuels and agriculture. CEDS estimates higher $CH_4$ emissions from fossil fuels while lower emissions from agriculture compared with EDGAR v4.3.2. A number of top-down emission inversion studies using surface and satellite observations have found that the EDGAR v4.3.2 (Maasakkers et al., 2019; Miller et al., 2019) and previous EDGAR versions (Alexe et al., 2015; Thompson et al., 2015; Turner et al., 2015; Pandey et al., 2016) overestimated the $CH_4$ emissions from coal production in China, likely due to the $CH_4$ emission factors for coal mining are too high in the region (Peng et al., 2016). A recent

bottom-up estimate suggested that Chinese coal mining $CH_4$ emissions have been decreasing since 2012 driven by the China's coal mine regulation (Sheng et al., 2019), but the interannual trend in Chinese coal emissions still has large uncertainties among studies (Miller et al., 2019; Sheng et al., 2019; Lu et al., 2021).

We also find that the interannual variability of OH concentrations can strongly affect the simulated $CH_4$ concentration trends.

Using interannually fixed OH concentrations, the model would overestimate the observed $CH_4$ growth since 2007 in China with both the EDGAR and CEDS anthropogenic emissions. The influence of a 10% increase in the global volume-weighted mean OH concentration (from $10.9\times10^5$ molecule $cm^{-3}$ to $12.0\times10^5$ molecule $cm^{-3}$) on the simulated Chinese $CH_4$ concentrations is equivalent to that of a 47 Tg $a^{-1}$ decrease in global $CH_4$ emissions. The use of interannual variability of OH provided by Zhao et al. (2019) improve the model simulated Chinese $CH_4$ concentrations and trends. However, large

discrepancies exist in the different model OH simulations that would lead to a large wide range ($>\pm30$ ppbv) of simulated $CH_4$ concentrations (Zhao et al., 2019). Despite these uncertainties, our study emphasizes the importance of emission changes in both domestic and foreign, anthropogenic and natural sources on the Chinese $CH_4$ concentration trends. Future work with more intensive $CH_4$ measurements covering the eastern China will help better assess the driving factors of Chinese $CH_4$ concentrations and recent growth.


**Data availability**

NOAA surface observations are available online (https://www.esrl.noaa.gov/gmd/dv/data/, last access: 5 May 2021). The GOSAT Proxy XCH4 data can be accessed through the Copernicus C3S Climate Data Store at https://cds.climate.copernicus.eu. The HIPPO data used in this study can be requested through

https://www.eol.ucar.edu/field_projects/hippo (last access: 5 May, 2021). The ATom data is available at https://daac.ornl.gov/cgi-bin/dsviewer.pl?ds_id=1581 (last access: 5 May, 2021). Modelling dataset can be accessed by contacting the corresponding author.



**Author contributions**

HYT and LZ designed the study. HYT conducted the modeling and data analyses with contributions from LZ, XL, YHZ and

BY. RJP and HB provided the GOSAT $CH_4$ data and contributed to the interpretation and discussion of its use in the study.
HYT and LZ wrote the paper with input from all authors.

**Competing interests**

The authors declare that they have no conflict of interest.

**Acknowledgments**

This work was supported by the National Natural Science Foundation of China (NSFC, 41475112, 41922037) and the
National Key Research and Development Program of China (2017YFC0210102). RJP and HB are funded via the UK
National Centre for Earth Observation (NE/N018079/1 and NE/R016518/1). We thank the Japanese Aerospace Exploration
Agency, National Institute for Environmental Studies, and the Ministry of Environment for the GOSAT data and their
continuous support as part of the Joint Research Agreement. This research used the ALICE High Performance Computing
Facility at the University of Leicester for the GOSAT retrievals. We thank Kathryn McKain and Steven C. Wofsy for
offering the $CH_4$ dataset from ATom campaigns.

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

## Tables and Figures

**Table 1. CH$_4$ measurements and GEOS-Chem model simulations at four NOAA surface sites over China.**

| Case | Resolution | Emission | OH | DSI (20.7ºN, 116.7ºE) | | LLN (23.5ºN, 120.9ºE) | | SDZ (40.7ºN, 117.1ºE) | | WLG (36.3ºN, 100.9ºE) | | |
| --- | --- | --- | --- | --- | --- | --- | --- | --- | --- | --- | --- | --- |
| | | | | 2010.03–2018.12 | | 2006.08–2018.12 | | 2009.09–2015.09 | | 2000.01–2006.12 | 2007.01–2018.12 | |
| | | | | Mean | Trend | Mean | Trend | Mean | Trend | Mean | Mean | Trend |
| Obs. | / | / | / | 1884.8 | 7.94 | 1851.7 | 7.03 | 1954.6 | 7.13 | 1832.4 | 1878.4 | 7.25 |
| Run1 | 4º×5º | EDGAR | Fixed | +0.4% | 8.74 (+0.8) | −3.3% | 9.63 (+2.6) | −2.9% | 13.33 (+6.2) | −5.5% | −3.6% | 9.35 (+2.1) |
| Run2 | 4º×5º | CEDS | Fixed | +4.4% | 11.94 (+4.0) | +0.4% | 13.23 (+6.2) | +2.3% | 18.03 (+10.9) | −3.8% | +0.2% | 12.85 (+5.6) |
| Run3 | 4º×5º | EDGAR | Varying | +0.7% | 4.24 (−3.7) | −2.5% | 4.83 (−2.2) | −2.1% | 8.23 (+1.1) | −2.6% | −2.9% | 4.55 (−2.7) |
| Run4 | 4º×5º | CEDS | Varying | +4.6% | 7.34 (−0.6) | +1.2% | 8.33 (+1.3) | +3.0% | 12.73 (+5.6) | −1.0% | +0.9% | 7.95 (+0.7) |
| Run5 | 4º×5º | CEDS | Varying (−10% over 2010–2018) | +8.0% | 19.14 (+11.2) | +3.5% | 18.53 (+11.5) | +5.0% | 25.93 (+18.8) | −1.0% | +3.4% | 18.85 (+11.6) |
| Run6 | 4º×5º | CEDS | Varying (+10% over 2010–2018) | +1.4% | −3.56 (−11.5) | −1.1% | −1.17 (−8.2) | +1.1% | 0.23 (−6.9) | −1.0% | −1.5% | −2.25 (−9.5) |
| Run7 | 4º×5º | CEDS (−50 Tg over 2010–2018) | Varying | +0.8% | −4.16 (−12.1) | −1.3% | −1.87 (−8.9) | +0.3% | −2.27 (−9.4) | −1.0% | −1.8% | −3.15 (−10.4) |
| GCE | 2º×2.5º | EDGAR | Varying | −2.0% | 3.54 (−4.4) | −1.5% | 4.63 (−2.4) | +1.8% | 9.73 (+2.6) | −2.6% | −2.8% | 4.65 (−2.6) |
| GCC | 2º×2.5º | CEDS | Varying | +1.6% | 6.44 (−1.5) | +1.6% | 8.13 (+1.1) | +6.5% | 14.73 (+7.6) | −1.6% | +0.3% | 7.95 (+0.7) |



**Table 2. CH$_4$ sources and sinks over China in 2000s and 2010s[a].**

| | | Saunois et al. (2020) | | | | Peng et al. (2016) | This study | | | |
|---|---|---|---|---|---|---|---|---|---|---|
| Time period | | 2000–2009 | | 2008–2017 | | 2000–2009 | 2000–2009 | | 2008–2017 | |
| Approach | | B-U | T-D | B-U | T-D | B-U | GCE | GCC | GCE | GCC |
| Sources (Tg a$^{-1}$) | Agriculture and waste | 27.1 (24.0–33.0) | 23.2 (10.4–28.3) | 29.7 (25.8–37.2) | 27.7 (10.9–34.7) | 22.9 | 33.3 | 25.3 | 36.8 | 28.2 |
| | Biomass and biofuel burning | 3.3 (1.8–5.0) | 3.6 (0.3–4.9) | 3.2 (1.3–5.1) | 3.7 (0.3–5.0) | 2.3 | 4.8 | 3.6 | 4.8 | 3.9 |
| | Fossil fuels | 17.9 (12.6–23.9) | 13.3 (7.4–31.0) | 26.1 (16.6–39.6) | 19.0 (8.0–35.6) | 12.8 | 15.8 | 23.4 | 22.8 | 38.4 |
| | Wetlands | 2.6 (0.9–9.3) | 6.0 (2.7–12.5) | 2.6 (0.8–9.2) | 5.2 (2.0–13.1) | / | 2.8 | | 2.8 | |
| | Other sources | / | 0.8 (0.6–1.6) | / | 0.8 (0.5–1.6) | / | 0.5 | 0.4 | 0.4 | 0.4 |
| Sinks (Tg a$^{-1}$) | Soils | / | 1.5 (0.8–2.0) | / | 1.8 (0.8–2.2) | / | 0.9 | | 0.9 | |
| | OH chemical loss | / | 8 | / | / | / | 6.3 | 6.4 | 6.7 | 6.9 |

[a] Bottom-up (B-U) and top-down (T-D) sources and soil uptake estimates of mean (range) values reported from Saunois et al., (2020) and
Peng et al., (2016). OH chemical loss estimates in 2000s are from Kirschke et al., (2013).






**Table 3. Sources contributing to the mean surface CH₄ concentration in China over 2007–2018ª**

| Concentration [%] | AGR | ENE | WST | RCO | BBN | WTL | SEE | TER | OTH | TOT |
|---|---|---|---|---|---|---|---|---|---|---|
| China | 3.4 | 4.1 | 2.0 | 0.9 | 0.1 | 0.7 | 0.1 | 0.3 | 0.0 | 11.4 |
| India | 3.6 | 0.4 | 0.9 | 0.3 | 0.0 | 0.5 | 0.0 | 0.1 | 0.0 | 6.0 |
| Rest Asia | 3.7 | 1.3 | 1.4 | 0.3 | 0.3 | 5.2 | 0.1 | 0.2 | 0.0 | 12.6 |
| Europe | 3.9 | 6.6 | 2.6 | 0.3 | 0.1 | 2.9 | 0.2 | 0.2 | 0.1 | 17.0 |
| Africa | 2,4 | 4.4 | 1.1 | 0.4 | 1.3 | 5.9 | 0.2 | 0.7 | 0.0 | 16.6 |
| North America | 2.2 | 3.0 | 1.6 | 0.1 | 0.1 | 6.6 | 0.2 | 0.2 | 0.1 | 13.9 |
| South America | 4.0 | 3.2 | 1.0 | 0.1 | 0.4 | 11.2 | 0.2 | 0.5 | 0.0 | 20.6 |
| Oceania | 0.7 | 0.2 | 0.2 | 0.0 | 0.2 | 0.3 | 0.0 | 0.1 | 0.0 | 1.8 |
| Rest World | 0.0 | 0.0 | 0.0 | 0.0 | 0.0 | 0.0 | 0.0 | 0.0 | 0.0 | 0.1 |
| Total | 24.0 | 23.1 | 10.9 | 2.5 | 2.6 | 33.4 | 1.0 | 2.4 | 0.3 | 100 |

ª Percentage contributions of CH₄ emissions from the CEDS sectors (Table S2) including agriculture (AGR), energy (ENE), wastewater (WST), residents (RCO), biomass burning (BBN), wetlands (WTL), seeps (SEE), termites (TER), and others (OTH) including industry (IND), transportation (TRA) and shipping (SHP), and from different regions (Fig. 2). Values are estimated using the tagged CH₄ tracer simulation.










**Table 4. Contributions of region- and sector-specific emissions to the surface CH₄ trends in China over 2007–2018ᵃ**

| Trend [ppbv a⁻¹] | AGR | ENE | WST | RCO | WTL | OTH | Total |
|---|---|---|---|---|---|---|---|
| China | 0.53 | 4.32 | 1.00 | −0.18 | −0.01 | 0.02 | 5.68 |
| India | 0.24 | 0.19 | 0.37 | 0.04 | 0.00 | 0.01 | 0.85 |
| Rest Asia | 0.33 | 0.61 | 0.62 | −0.02 | −0.19 | −0.01 | 1.35 |
| Europe | −1.58 | −1.59 | −0.12 | −0.18 | −0.03 | −0.06 | −3.56 |
| Africa | 0.59 | 2.03 | 0.55 | 0.11 | 0.01 | −0.08 | 3.20 |
| North America | 0.03 | −0.20 | −0.18 | −0.03 | −0.11 | −0.04 | −0.53 |
| South America | 0.64 | 1.08 | 0.32 | −0.01 | −0.60 | −0.04 | 1.39 |
| Oceania | −0.03 | 0.04 | −0.02 | −0.01 | -0.03 | −0.02 | −0.07 |
| Rest World | 0.00 | 0.00 | 0.00 | 0.00 | 0.00 | 0.00 | 0.01 |
| Total | 0.75 | 6.49 | 2.54 | −0.28 | −0.95 | −0.22 | 8.32 |

ᵃ Contributions of CH₄ sources from different CEDS sectors and from different regions to the mean surface trends in China over 2007–2018. The CEDS sectors include agriculture (AGR), energy (ENE), wastewater (WST), residents (RCO), wetland (WTL), and others (OTH) combining industry, transportation, shipping, biomass burning, seeps, and termites (Table S2). Values are estimated using the tagged CH₄ tracer simulation as described in the text.






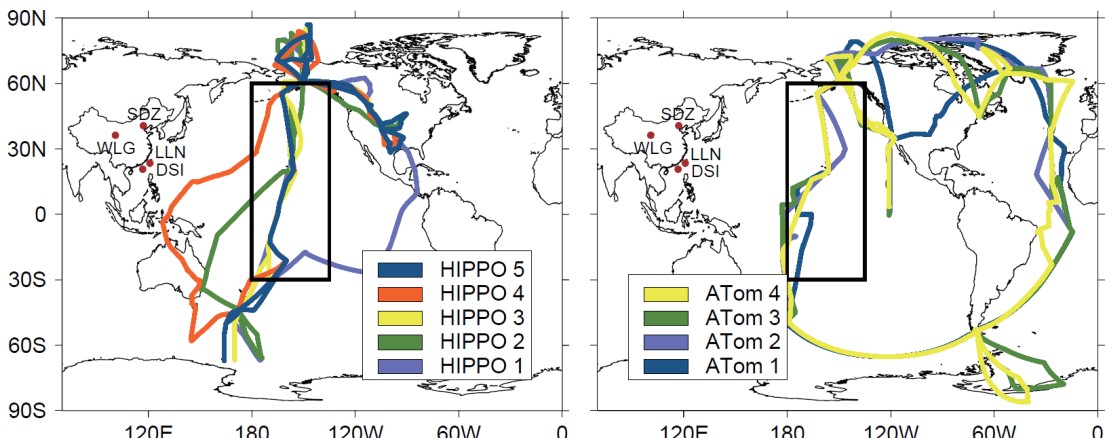

**Figure 1.** Locations of background surface sites in China and aircraft flight tracks of HIPPO and ATom campaigns.


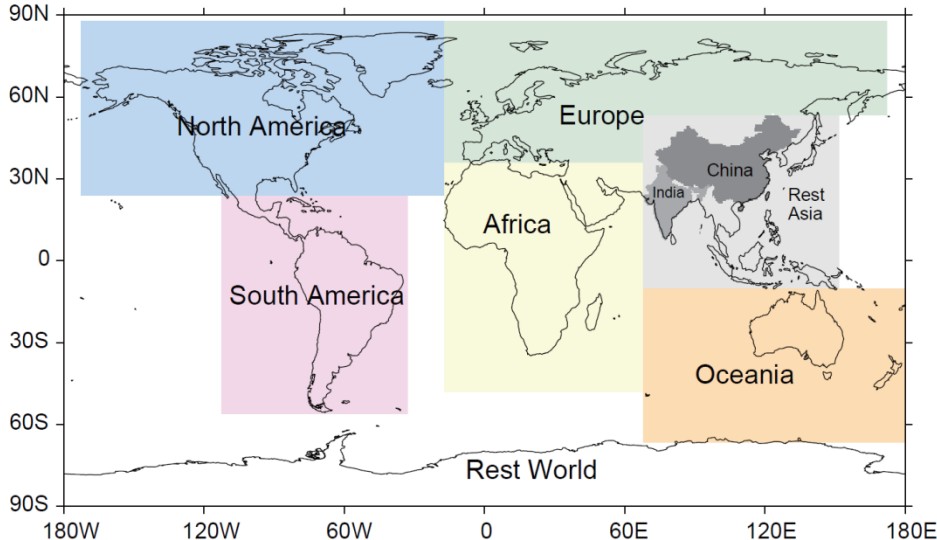

**Figure 2.** Regions defined in the tagged $CH_4$ tracer simulation. The regions are North America (NA), South America (SA), Europe (EU), Africa (AF), Oceania (OC), China (CHN), India (IND) and Rest Asia (RtAS). All other areas are included in the rest of the world (RW) region.




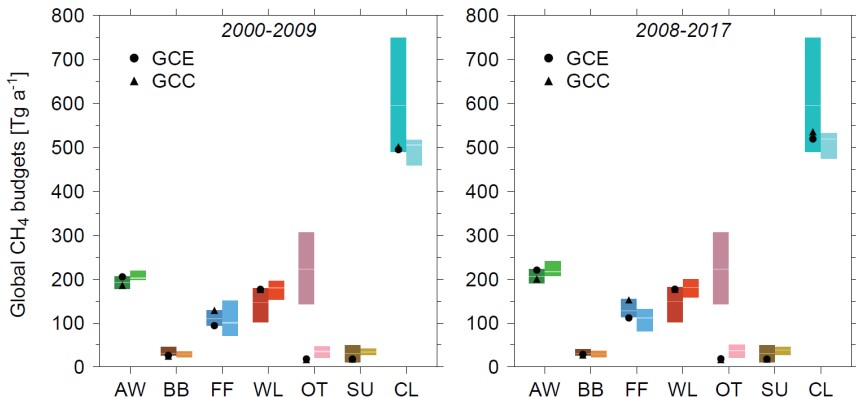

**Figure 3.** Global CH$_4$ budgets from main source categories and sinks for the 2000–2009 (2000s) and 2008–2017 (2010s) periods. Categories are grouped based on Table S2 including emissions from agriculture and waste (AW), fossil fuels (FF), wetlands (WL), biomass burning (BB), and others (OT), and sinks due to soil uptake (SU) and chemical loss (CL). The bar charts show bottom-up (dark-colored bars) and top-down (light-colored bars) estimates in previous studies as summarized by Saunois et al. (2020). The global CH$_4$ sources and sinks in the GCE (black circles) and GCC (black triangles) model simulations are also shown. Table S1 summarizes the values presented in the figure.





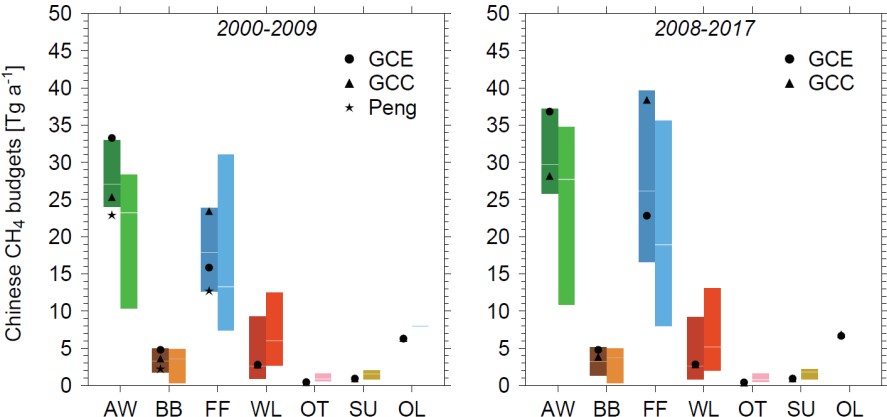

**Figure 4.** Similar to Fig. 3, but for CH$_4$ sources and sinks over China averaged for the 2000–2009 and 2008–2017 periods. The bar charts show previous Chinese bottom-up (dark-colored bars on the left) and top-down (light-colored bars on the right) estimates as summarized by Saunois et al. (2020) and Kirschke et al. (2013), and are compared with model results in the GCE (black circles) and GCC (black triangles) simulations. The bottom-up estimates of 2000–2009 mean Chinese CH$_4$ emissions by Peng et al. (2016) are also shown as black stars. Values presented in the figure are summarized in Table 2.



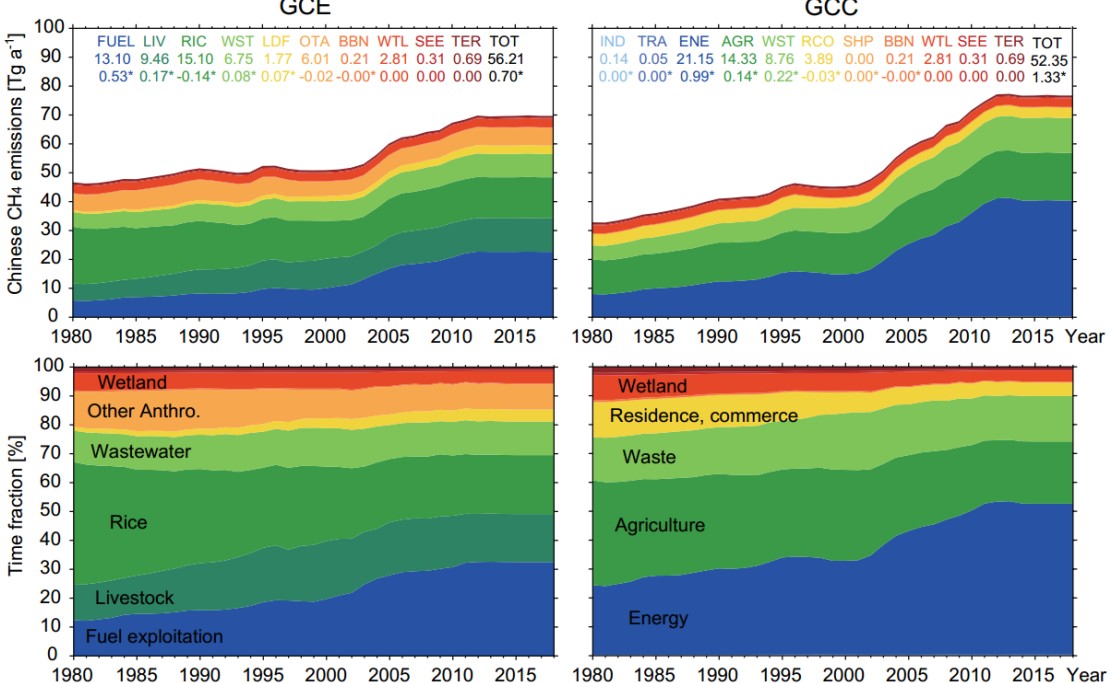

**Figure 5.** Time series of annual Chinese CH$_4$ emissions from different sectors (top panels) and their percentage contributions (bottom panels) in the GCE (left panels) and GCC (right panels) model simulations during the period of 1980–2018. The emission sectors and their abbreviations are listed in Table S2. Annual mean emission totals and trends over 1980–2018 (with asterisks denoting the statistical significance of p-value < 0.05) are shown inset.





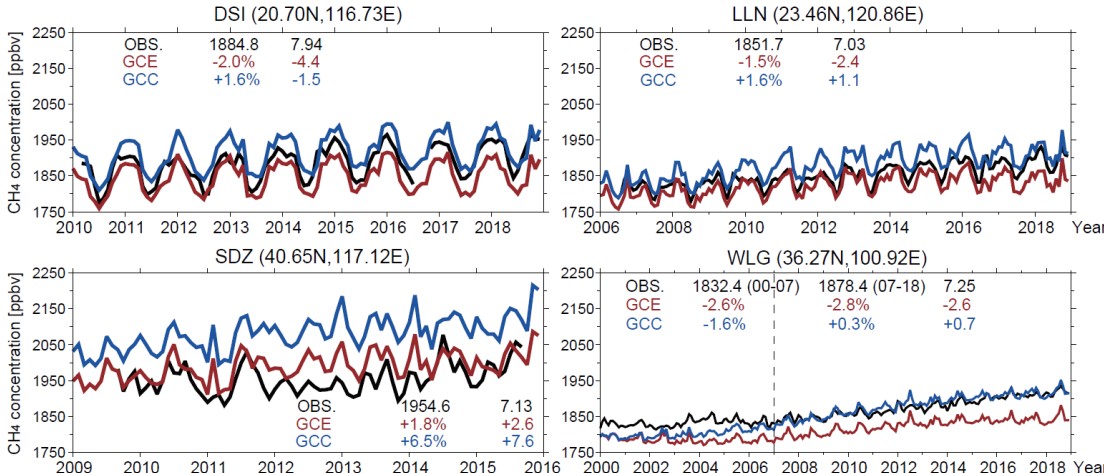

**Figure 6.** Comparison of GCE (red lines) and GCC (blue lines) simulated monthly mean CH₄ concentrations with NOAA in situ observations (black lines) in China. The observed mean concentrations (in unit of ppbv), trends (ppbv a⁻¹), and corresponding model biases are shown inset.


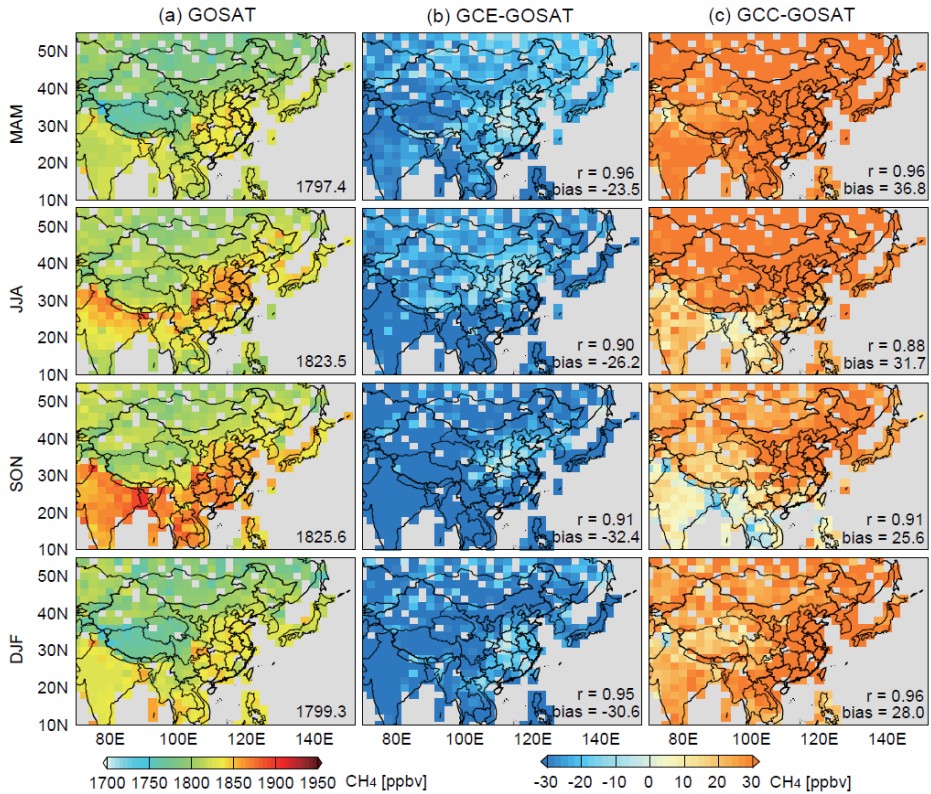

**Figure 7.** 2010–2017 seasonal mean GOSAT observed and model simulated atmospheric $CH_4$ concentrations over Asia. Both observations and the GEOS-Chem model simulations (GCE and GCC) are regrided to the 2º×2.5º model resolution. The model results are then applied with satellite averaging kernels. The middle and right panels show, respectively, GCE minus GOSAT and GCC minus GOSAT differences. The observed mean atmospheric $CH_4$ concentration, GOSAT vs. model correlation coefficients (*r*), and mean model biases over China are shown inset. The seasonal means are averages of March-April-May (MAM), June-July-August (JJA), September-October-November (SON), and December-January-February (DJF).





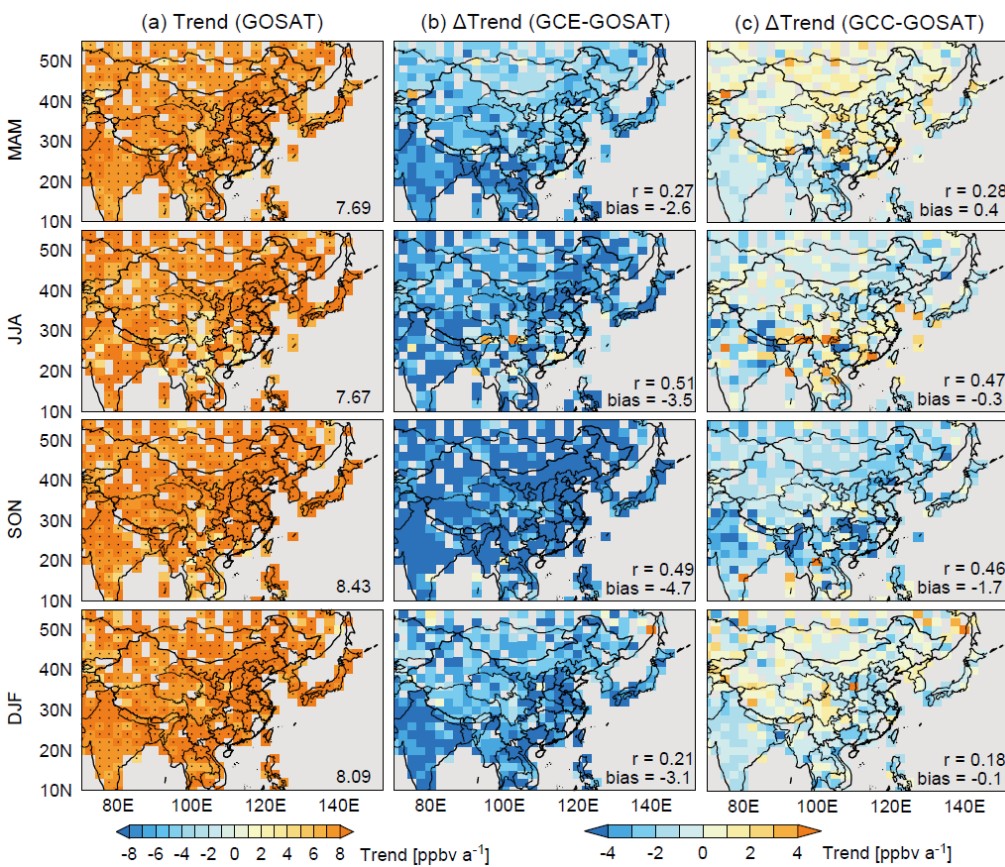

**Figure 8.** The same as Fig. 7, but for seasonal mean trends in atmospheric CH$_4$ concentrations over 2010–2017.





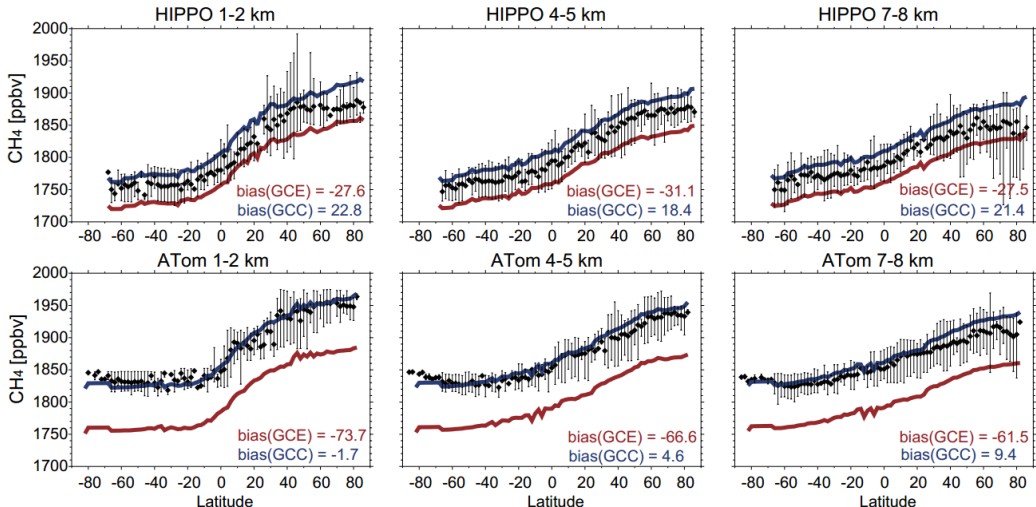

**Figure 9.** HIPPO and ATom aircraft measured latitudinal gradients of $CH_4$ concentrations. Measurements from HIPPO (top panels) and ATom (bottom panels) flights are averaged in 2° latitude bins and at three altitude levels (left: 1–2 km; middle: 4–5 km; and right: 7–8 km). The black symbols and bars represent the mean values and ranges for each bin. The corresponding model results from the GCE (red lines) and GCC (blue lines) simulations are also shown, and the values inset present the mean model biases relative to aircraft measurements.





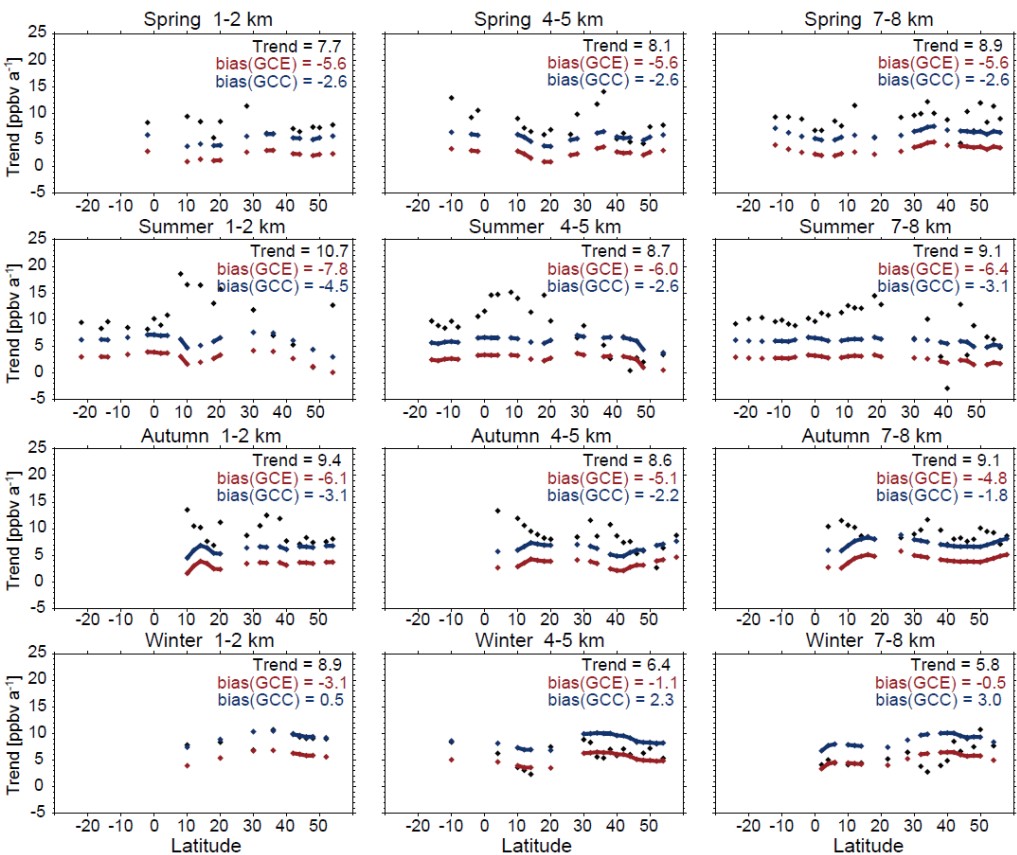

**Figure 10.** Comparisons of simulated CH$_4$ trends in GCE (red dots) and GCC (blue dots) against aircraft observation trends (black symbols) in four seasons (spring: March-April-May; summer: August; autumn: October-November; winter: January-February). All observations and model results sampled along the flight tracks are averaged in 2° latitude bins and at three altitude levels (left: 1–2 km; middle: 4–5 km; and right: 7–8 km). Mean observed CH$_4$ trends and model biases are shown inset.



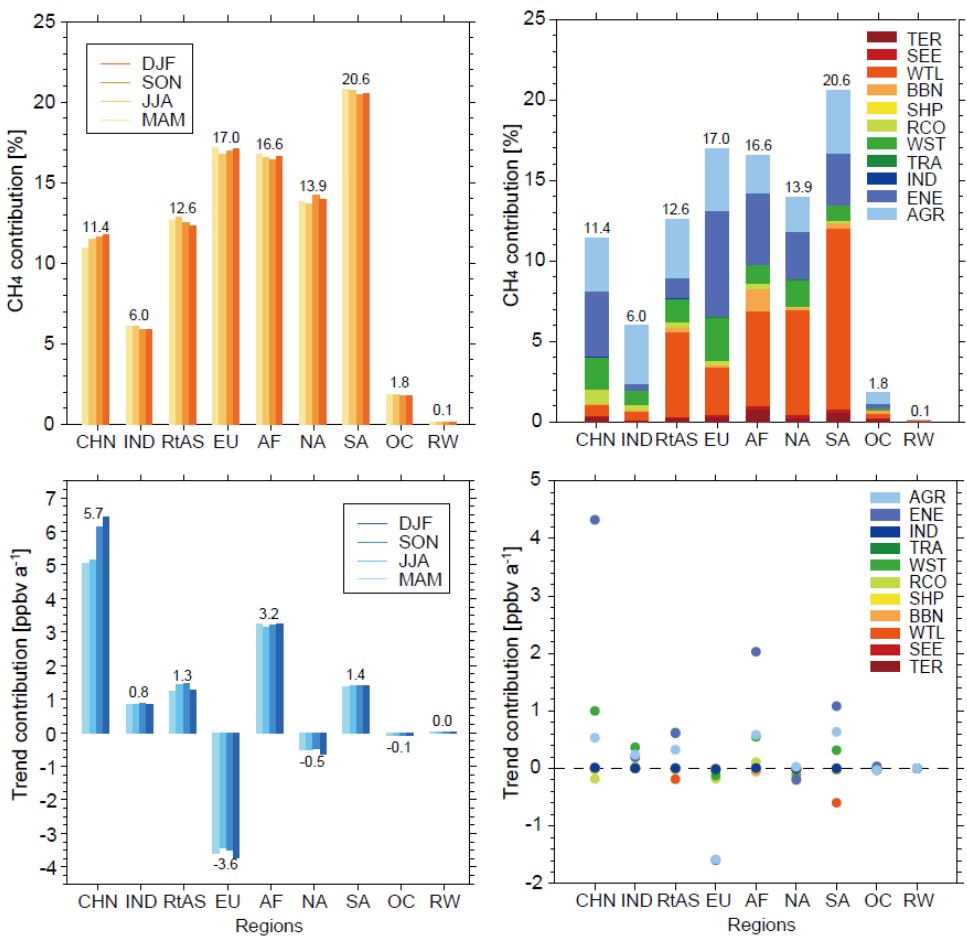

755

**Figure 11.** Contributions of CH$_4$ emissions from different regions and different source sectors on the mean surface CH$_4$ concentrations (top panels) and trends (bottom panels) in China over 2007–2018. The left panels show region-specific source contributions for different seasons and the right panels show region- and sector-specific contributions for the annual values. Source contributions are estimated using the tagged CH$_4$ tracers accounting for emission sources from agriculture (AGR), energy (ENE), industry (IND), transportation (TRA), residents (RCO), wastewater (WST), shipping (SHP), biomass burning (BBN), wetlands (WTL), seeps (SEE) and termites (TER) sectors and from nine regions (Africa, China, Europe, India, Asia excluding China and India, Oceania, South America, North America and the rest of the world).

760





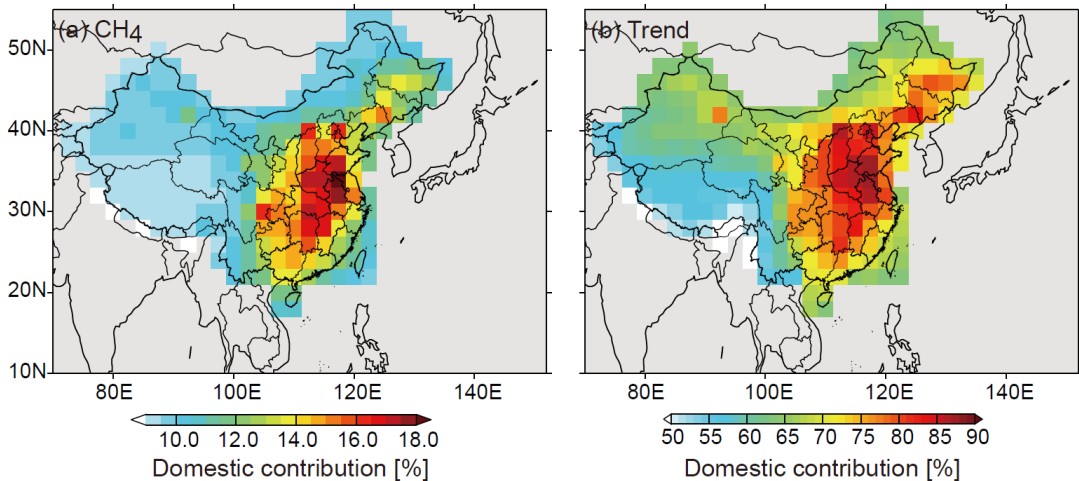

765    **Figure 12.** Spatial distributions of Chinese domestic emission contributions in percentage on CH$_4$ surface concentrations (a) and trends (b) in 2007–2018 over China. The percentage contributions are estimated by summing up all the Chinese tagged CH$_4$ tracers divided by the total CH$_4$ tracers in the tagged simulation.