# Peer review of "An integrated analysis of contemporary methane emissions and concentration trends over China using in situ, satellite observations, and model simulations"

_Atmospheric Chemistry and Physics, 2021_

## Referee Comment (RC2)

This work attempts to attribute the sources contributing to the atmospheric CH4 mixing ratio and their trends in China using the GEOS-Chem model simulations driven by two commonly used global anthropogenic emission inventories. It uses in-situ and satellite observations of CH4 mixing ratios to explain the model results. Study also performs sensitivity test with OH to reproduce observed CH4 mixing ratios and trends over China.

The discussion on the differences between the model results and observations is not sufficient. Authors can address some missing pieces of information or address some limitations in the results shown. I recommend the manuscript send back for the major revisions with following major/minor comments:

Major comments:

The study claims "model simulation using the CEDS inventory and interannually varying OH levels can best reproduce observed CH4 mixing ratios and trends over China". I don't agree to a certain extent.

First of all, EDGAR v4.3.2 provides global emission estimates, at source-sector level, for the historic period from 1970 until 2012. How did the author estimate the EDGAR emissions beyond 2012? It appears the emissions are extrapolated (?) till 2018 for this study. Similarly, in case of CEDS inventory the emission estimates are during 1970-2014. How does the emissions are calculated beyond 2014 in this case too?

In Figure 6, over 'DSI' and 'LLN', simulations from both inventories are comparable at least till the year 2016 (it appears that, trends are affected by later years simulations for GCE). Over 'SDZ', EDGAR performs better than CEDS, however, over 'WLD', CEDS is better than EDGAR. Overall, these results are not very conclusive to say CEDS is better.

In Figure 7-8, the trend correlations for model simulations using EDGAR and CEDS with GOSAT are not significantly different.

In Figure 9, it appears for HIPPO observations, both simulations (performed using EDGAR and CEDS emission inventory) are within observed standard deviation. But for ATOM observations, CEDS inventory performs better than EDGAR. One reason to me is EDGAR extrapolated(?) emissions are used for the model simulations comparison with ATOM observations, whereas, in case of HIPPO observations actual EDGAR emission estimates are used. So, this Figure is also not very conclusive to say CEDS inventory is better, moreover, almost all the observations from HIPPO and ATOMS are over Pacific and American continent.

Figure 10, mixes both aircraft observations, which is not correct in my opinion. This figure is confusing.

Another issue is the source attribution of CH4. The attribution of CH4 sources with tagged tracer needs more evidences. The source contribution should be provided along with confidence interval. Is there any relevant study to support this analysis for CH4?

Minor comments:
Some places in the manuscript authors use 'Fig.' and somewhere 'Figure'. Please use uniform convention.

Line 55; please add a reference after "a lifetime of 9.14 (±10%) years"

Fig2: How do you define the regions for tagged CH4 tracer simulations?

In Figure 6-10, please mention the model configure and OH field configuration used for simulations in the caption for better clarity.

Fig10: Legends needs to be adjusted properly.

---

## Author Response (AR1)

**Response letter for acp-2021-464**
**"An integrated analysis of contemporary methane emissions and concentration trends over China using in situ, satellite observations, and model simulations"**

**Dear Editor Tim Butler,**

**Thank you very much for handling our manuscript. Please find our point-to-point responses to three reviewers' comments below. We thank the three reviewers for their thoughtful comments and each comment has been implemented in the revised manuscript.**

**Sincerely,**
**Lin Zhang, Haiyue Tan, et al.**

- - - - - - - - - - - - - - - - - - - - - - - - - - - - - - - - - - - - - -

**Reviewer #1**
**Comment [1-1]:** The manuscript describes the recent methane budgets and concentrations over China, and contains comparisons and analyses of the model results from a global chemical transport model with three observation datasets. The authors elucidate the contributions of region-
sector-specific methane emissions to methane concentrations and trends which allow to better diagnose and understand the drivers of methane changes in China. The topic of the manuscript is certainly within the scope of ACP. Overall, the manuscript is well written and easy to follow, so it can be accepted after a minor revision.
**Response [1-1]:** We sincerely thank the reviewer for the positive and valuable comments, and
time spent reviewing the manuscript. The revised manuscript has implemented all of them. Please see our responses to each comment below.

**Comment [1-2]:** In Section 2.3, the GEOS-Chem model setup is described. But I cannot find the description about how long the simulations spinup or the statement about initial methane
concentrations. Additional brief information about how it conducted would be welcomed.
**Response [1-2]:** Thank you for pointing it out. We have tested that changes in the initial $CH_4$ conditions in January 1980 would not affect simulation results after January 2000, supporting a spin-up time of 20 years. We now add the following information in Section 2.3: "All the simulations are initiated in the year 1980 and we focus on the model results in the period of 2007–
2018. We find that changes in the initial $CH_4$ conditions in January 1980 would not affect simulation results after January 2000, indicating that a spin-up time of over 20 years is sufficient for our analyses."

**Comment [1-3]:** The description of $CH_4$ mixing ratio or concentration should be consistent. The
former is used in the abstract and the latter in most other parts. The "mixing ratio" often collocates with the unit of "ppbv" such as methane in this paper and "concentration" with "molec/cm$^3$" such as OH.
**Response [1-3]:** We now change in the text "$CH_4$ concentrations" into "$CH_4$ mixing ratios" following the suggestion except "$CH_4$ concentration trends" in the title for keeping it concise.

**Comment [2-1]:** The authors integrated emission inventories, GEOS-Chem simulations, in-situ, and GOSAT satellite retrievals to investigate $CH_4$ concentrations, sources, and sinks over China. Such an analysis is very important because $CH_4$ is the second most important GHG and China is the largest emitter of anthropogenic $CH_4$ in the world. However, we lack a comprehensive study to focus on China's methane concentrations and budget at present. This study is a good first step, and I recommend this paper for publication in ACP.

**Response [2-1]:** We sincerely thank the reviewer for the positive and valuable comments, and time spent reviewing the manuscript. The revised manuscript has implemented all of them. Please see our responses to each comment below.

**Comment [2-2]:** My main suggestion for the authors is that they can consider including the TCCON $XCH_4$ data to evaluate their GEOS-Chem simulations as well.

**Response [2-2]:** Thank you for pointing it out. We now add evaluations with column $CH_4$

measurements at six TCCON sites in Asia. The figure is added in the supplement as Figure S3 and shown below. In addition, we add the following text in the Section 3.2: "Further evaluations of the two model simulations with $CH_4$ column mixing ratio measurements (since 2011) at six TCCON sites in Asia (Wunch et al., 2011) show similar results, with small biases of 0.2%–1.0% in $CH_4$ mixing ratios for GCC and negative biases of 2.6%–3.7% for GCE (Fig. S3). This again reflects the higher Chinese $CH_4$ emission estimates in years around 2012 in CEDS than EDGAR, which then affect the model simulations afterwards by using their emissions of the latest available years."

[Figure]

**Figure S3.** Comparison of GCE (red lines) and GCC (blue lines) simulated monthly results with TCCON
observations of column $CH_4$ mixing ratios (black lines) in Asia. The observed mean mixing ratios (ppbv), trends at the two sites with more than 7-year measurements (ppbv $a^{-1}$), and corresponding model biases are shown inset. Locations of the six measurement sites are shown in Fig. 1.

**Comment [2-3]:** Besides, I suggest adding more figure legends to clarify that the global and
regional $CH_4$ budgets (except those from GCE and GCC) and China's $CH_4$ emissions data plotted in Figs. 3-5 are derived from previous literature, not the estimates of this study.

**Response [2-3]:** Thank you for the comment. We have now revised the legends of Figures 3 and 4 by adding dark-colored and light-colored bars to represent the ranges estimated from Saunois et al. (2020).

[Figure]

**Figure 3.** Global CH₄ budgets from main source categories and sinks for the 2000–2009 (2000s) and 2008–2017 (2010s) periods. Categories are grouped based on Table S2 including emissions from agriculture and waste (AW), fossil fuels (FF), wetlands (WL), biomass burning (BB), and others (OT), and sinks due to soil uptake (SU) and
chemical loss (CL). The bar charts show bottom-up (dark-colored bars) and top-down (light-colored bars) estimates in previous studies as summarized by Saunois et al. (2020). The global CH₄ sources and sinks in the GCE (black circles) and GCC (black triangles) model simulations are also shown. Table S1 summarizes the values presented in the figure.

[Figure]

**Figure 4.** Similar to Fig. 3, but for CH₄ sources and sinks over China averaged for the 2000–2009 and 2008–2017 periods. The bar charts show previous Chinese bottom-up (dark-colored bars on the left) and top-down (light-colored bars on the right) estimates as summarized by Saunois et al. (2020) and Kirschke et al. (2013), and are compared with model results in the GCE (black circles) and GCC (black triangles) simulations. The bottom-up estimates of
2000–2009 mean Chinese CH₄ emissions by Peng et al. (2016) are also shown as black stars. Values presented in the figure are summarized in Table 2.

**Reference**

Saunois, M., Stavert, A. R., et al.: The Global Methane Budget 2000–2017, Earth System Science Data,
12, 1561-1623, 10.5194/essd-12-1561-2020, 2020.

Wunch, D., Toon, G. C., Blavier, J. F., Washenfelder, R. A., Notholt, J., Connor, B. J., Griffith, D. W.,
    Sherlock, V., and Wennberg, P. O.: The total carbon column observing network, Philos Trans A Math
    Phys Eng Sci, 369, 2087-2112, 10.1098/rsta.2010.0240, 2011.

**Comment [3-1]:** This work attempts to attribute the sources contributing to the atmospheric $CH_4$ mixing ratio and their trends in China using the GEOS-Chem model simulations driven by two commonly used global anthropogenic emission inventories. It uses in-situ and satellite observations of $CH_4$ mixing ratios to explain the model results. Study also performs sensitivity test with OH to reproduce observed $CH_4$ mixing ratios and trends over China.

The discussion on the differences between the model results and observations is not sufficient. Authors can address some missing pieces of information or address some limitations in the results shown. I recommend the manuscript send back for the major revisions with following major/minor comments.

**Response [3-1]:** We sincerely thank the reviewer for the constructive comments, and time spent reviewing the manuscript. Each comment has been implemented in the revised manuscript.

The main comments raised by the reviewer concern about whether our model evaluations support the use of CEDS v2017-05-18 emissions rather than EDGAR v4.3.2 for simulating the $CH_4$ trends over China in the time period of 2007-2018. This study does not attempt to provide a conclusive judgement that the $CH_4$ emission estimates in CEDS are more accurate than those in EDGAR. Instead based on analyses of a series of global model simulations accounting for the interannually varying OH and considering their latest available years (2012 for EDGAR and 2014 for CEDS), we find that the rising $CH_4$ levels over China over 2007-2018 can be better captured in the model simulation with the CEDS inventory. This provides an important constraint on the trends of $CH_4$ emissions over China. The comments from the reviewer are valuable and help us better interpret the model simulations. Please see our point-to-point responses below.

**Comment [3-2]:** Major comments: The study claims "model simulation using the CEDS inventory and interannually varying OH levels can best reproduce observed $CH_4$ mixing ratios and trends over China". I don't agree to a certain extent.

First of all, EDGAR v4.3.2 provides global emission estimates, at source-sector level, for the historic period from 1970 until 2012. How did the author estimate the EDGAR emissions beyond 2012? It appears the emissions are extrapolated (?) till 2018 for this study. Similarly, in case of CEDS inventory the emission estimates are during 1970-2014. How does the emissions are calculated beyond 2014 in this case too?

**Response [3-2]:** Thank you for pointing it out. For both EDGAR and CEDS, we did not extrapolate them, and instead as we now state in Section 2.3: "For all the datasets of emissions (using EDGAR or CEDS) and sinks as described above, the closest available year will be used for simulation years beyond their available time ranges." This is mainly because of the large uncertainties in global $CH_4$ emission estimates (Saunois et al., 2020) as well as the slow trends of emissions in China after 2010 compared to 2000s (Sheng et al., 2021; Liu et al., 2021). As methane has a lifetime of about 9 years, the changes of $CH_4$ mixing ratios after 2012 are strongly affected by its emissions before, which drives the model differences with EDGAR vs. with CEDS.

We now state here in Section 2.3: "the closest available year will be used for simulation years beyond their available time ranges as recent studies suggested weak trends in Chinese $CH_4$ emissions after 2010 (Sheng et al., 2021; Liu et al., 2021). Since $CH_4$ has a long lifetime of about 9 years, model results in the later years (e.g., after 2012 for EDGAR and after 2014 for CEDS) are strongly affected
       by the emissions in earlier years".

       **Comment [3-3]:** In Figure 6, over 'DSI' and 'LLN', simulations from both inventories are
       comparable at least till the year 2016 (it appears that, trends are affected by later years simulations
for GCE). Over 'SDZ', EDGAR performs better than CEDS, however, over 'WLD', CEDS is better
       than EDGAR. Overall, these results are not very conclusive to say CEDS is better.
       **Response [3-3]:** As we discussed above, simulated $CH_4$ levels at these Chinese sites after the year
       2016 would still be strongly affected by emissions in earlier years. The high bias at SDZ in the
       CEDS model simulation likely suggest that regional $CH_4$ emissions around this site (i.e., North
China) are too high in CEDS.

       We now state in the Section 3.2, "These results can be explained by the higher $CH_4$ emission
       estimates and increases in CEDS than EDGAR since 2007, and may also reflect the regional $CH_4$
       emissions around SDZ (i.e., North China) are too high in CEDS. Further evaluations of the two
model simulations with $CH_4$ column mixing ratio measurements (since 2011) at six TCCON sites
       in Asia (Wunch et al., 2011) show similar results, with small biases of 0.2%–1.0% in $CH_4$ mixing
       ratios for GCC and negative biases of 2.6%–3.7% for GCE (Fig. S3). This again reflects the higher
       Chinese $CH_4$ emission estimates in years around 2012 in CEDS than EDGAR, which then affect the
       model simulations afterwards by using their emissions of the latest available years."

       **Comment [3-4]:** In Figure 7-8, the trend correlations for model simulations using EDGAR and
       CEDS with GOSAT are not significantly different.
       **Response [3-4]:** Both correlation coefficients (r) and mean biases over China between model
       simulations and GOSAT are presented in Figs. 7 and 8. The correlation coefficients show the
similarity of spatial distributions between observed $CH_4$ mixing ratio (or trend) and model results,
       and are not significantly different for the two model simulations, however, the mean biases indicate
       that model results with EDGAR underestimate the GOSAT observed trends in $CH_4$ mixing ratios
       over China.

We now state in the text: "As for the $CH_4$ trends during 2010–2017 over China, both GCC and GCE
       show similar spatial patterns as those observed by GOSAT with moderate correlations of 0.2–0.5,
       while GCC model results have smaller biases of $-1.7$–0.4 ppbv $a^{-1}$, compared to GCE results that
       underestimate the trends by 2.6–4.7 ppbv $a^{-1}$."

**Comment [3-5]:** In Figure 9, it appears for HIPPO observations, both simulations (performed using
       EDGAR and CEDS emission inventory) are within observed standard deviation. But for ATOM
       observations, CEDS inventory performs better than EDGAR. One reason to me is EDGAR
       extrapolated (?) emissions are used for the model simulations comparison with ATOM observations,
       whereas, in case of HIPPO observations actual EDGAR emission estimates are used. So, this Figure
is also not very conclusive to say CEDS inventory is better, moreover, almost all the observations
       from HIPPO and ATOMS are over Pacific and American continent.
       **Response [3-5]:** Thanks for the comment. As responded above, both EDGAR and CEDS emissions
       were not extrapolated, and we do not think this would affect our analyses much due to the long lifetime of CH$_4$. We suggest as stated in the text that "changes in the model bias for the comparisons with HIPPO and ATom measurements reflect their simulated trends in the CH$_4$ mixing ratios". HIPPO and ATom are two aircraft campaigns well designed for measuring global-scale atmospheric composition. The measurements over Pacific account for influences of Asian outflows, and thus can be applied for evaluating model CH$_4$ simulations with information on Asian CH$_4$ sources. Please also see our response to the next comment.

**Comment [3-6]:** Figure 10, mixes both aircraft observations, which is not correct in my opinion. This figure is confusing.

**Response [3-6]:** Thanks for raising the concern, yet we think this comparison is valid for long lived tracers such as CH$_4$. Both HIPPO and ATom campaigns provide measurements over the Pacific region in all four seasons and with extensive vertical profiling, supporting the analyses shown in Figure 10.

We now state in Section 2.1, "Both campaigns provide global-scale measurements of atmospheric composition in all seasons, and conduct continuous profiling between ~0.15 km and 8.5 km altitude with many profiles extending to nearly 14 km". We state in Section 3.2, "Since both HIPPO (2009–2011) and ATom (2016–2018) provide measurements over the Pacific (black box in Fig. 1), we calculate the differences between HIPPO and ATom measurements as the observed CH$_4$ concentration trends over this region, and these trends also largely reflect the influences from upwind Asian CH$_4$ sources and levels". In addition, we have revised the legends of Figure 10 for clarification following one minor comment below.

**Comment [3-7]:** Another issue is the source attribution of CH$_4$. The attribution of CH$_4$ sources with tagged tracer needs more evidences. The source contribution should be provided along with confidence interval. Is there any relevant study to support this analysis for CH$_4$?

**Response [3-7]:** Thanks for the comment. The tagged tracer approach has been applied in a number of studies. We now stated in Section 2.3, "The tagged CH$_4$ tracer approach has been recently applied in GEOS-Chem to quantify source contributions in U.S. Midwest (Yu et al., 2021) and GFDL-AM4.1 with focuses on the global CH$_4$ budget (He et al., 2020)". We have now added error-bars on the left panels of Figure 11 to present the standard deviations of contributions from different regions as a metric of confidence level. The values are up to 11% for the contributions to CH$_4$ mixing ratios and up to 0.4 ppbv a$^{-1}$ to the trends over China. We state in the text "We find strong spatial variation in the contribution values over different regions of China with standard deviations up to 11% for the contributions to CH$_4$ mixing ratios and up to 0.4 ppbv a$^{-1}$ to the trends (Fig. 11)."

We also acknowledge that the source attribution results can heavily rely on the regional and sectorial estimates of underlying CH$_4$ emissions. We have stated in the last section that "It shall be noted that our source attribution results can be biased by the use of CEDS and the uncertainty in the interannual variations of OH levels", and discussed the uncertainties in this paragraph.

[Figure]

**Figure 11.** Contributions of CH$_4$ emissions from different regions and different source sectors on the mean surface CH$_4$ mixing ratios (top panels) and trends (bottom panels) in China over 2007–2018. The left panels show region-specific source contributions for different seasons and error bars are standard deviations denoting spatial variation of contributions over China. The right panels show region- and sector-specific contributions for the annual values. Source contributions are estimated using the tagged CH$_4$ tracers accounting for emission sources from agriculture (AGR), energy (ENE), industry (IND), transportation (TRA), residents (RCO), wastewater (WST), shipping (SHP), biomass burning (BBN), wetlands (WTL), seeps (SEE) and termites (TER) sectors and from nine regions (Africa, China, Europe, India, Asia excluding China and India, Oceania, South America, North America and the rest of the world).

**Comment [3-8]:** Minor comments: Some places in the manuscript authors use 'Fig.' and somewhere 'Figure'. Please use uniform convention.

**Response [3-8]:** Thank you for pointing it out. The uses of 'Fig.' and 'Figure' have followed the request of ACP format, i.e., the abbreviation "Fig." should be used when it appears in running text and should be followed by a number unless it comes at the beginning of a sentence.

**Comment [3-9]:** Line 55; please add a reference after "a lifetime of 9.14 (±10%) years"

**Response [3-9]:** We now add the reference in the text: "Over 90% of atmospheric CH$_4$ is lost via oxidation by OH in the troposphere, leading to a lifetime of 9.14 (±10%) years against this sink (IPCC, 2013)."

**Comment [3-10]:** Fig2: How do you define the regions for tagged CH$_4$ tracer simulations?

**Response [3-10]:** The nine regions (China, India, Europe, South America, North America, Africa, Oceania, Rest Asia and the rest world) are defined mainly following Bey et al. (2001) with some modifications. We now state in the text: "The regions used for the tagged simulation are shown in Fig. 2, mainly based on Bey et al. (2001) with additional tagged regions for China and India in Asia."

**Comment [3-11]:** In Figure 6-10, please mention the model configure and OH field configuration used for simulations in the caption for better clarity.

**Response [3-11]:** We now clarify in the caption of Figs. 6-10: "GCE (with EDGAR anthropogenic emissions and interannually varying OH; red lines) and GCC (with CEDS and interannually varying

OH; blue lines)"

**Comment [3-12]:** Fig10: Legends needs to be adjusted properly.

**Response [3-12]:** Thanks for pointing it out. The legends in Fig. 10 have been adjusted from "Trend = …", "bias (GCE) = …", "bias (GCC) = …" to "Obs. …", "GCE …", "GCC …" to better present the mean values.

[Figure]

**Figure 10.** Comparisons of simulated CH$_4$ trends in GCE (with EDGAR anthropogenic emissions and interannually varying OH; red dots) and GCC (with CEDS and interannually varying OH; blue dots) against aircraft observation trends (black symbols) in four seasons (spring: March-April-May; summer: August; autumn: October-November; winter: January-February). All observations and model results sampled along the flight tracks are averaged in 2° latitude bins and at three altitude levels (left: 1–2 km; middle: 4–5 km; and right: 7–8 km). Mean observed and simulated CH$_4$ trends are shown inset.

**Reference**

Bey, I., Jacob, D. J., Yantosca, R. M., Logan, J. A., Field, B. D., Fiore, A. M., Li, Q. B., Liu, H. G. Y., Mickley, L. J., and Schultz, M. G.: Global modeling of tropospheric chemistry with assimilated meteorology: Model description and evaluation, J Geophys Res-Atmos, 106, 23073-23095, Doi 10.1029/2001jd000807, 2001.

He, J., Naik, V., Horowitz, L. W., Dlugokencky, E., and Thoning, K.: Investigation of the global methane budget over 1980–2017 using GFDL-AM4.1, Atmospheric Chemistry and Physics, 20, 805-827, 10.5194/acp-20-805-2020, 2020.

IPCC: Carbon and other biogeochemical cycles, in: Climate Change 2013: The Physical Science Basis. Contribution of Working Group I to the Fifth Assessment Report of the Intergovernmental Panel on Climate Change, edited by: Stocker, T. F., Qin, D., Plattner, G.-K., Tignor, M., Allen, S. K., Boschung, J., Nauels, A., Xia, Y., Bex, V., and Midgley, P. M., Cambridge University Press, Cambridge, United Kingdom and New York, NY, USA, 465-570, 2013.

Liu, G., Peng, S., Lin, X., Ciais, P., Li, X., Xi, Y., Lu, Z., Chang, J., Saunois, M., Wu, Y., Patra, P.,

Chandra, N., Zeng, H., and Piao, S.: Recent Slowdown of Anthropogenic Methane Emissions in China Driven by Stabilized Coal Production, Environmental Science & Technology Letters, 8, 739-746, 10.1021/acs.estlett.1c00463, 2021.

Saunois, M., Stavert, A. R., Poulter, B., Bousquet, P., Canadell, J. G., Jackson, R. B., Raymond, P. A., Dlugokencky, E. J., Houweling, S., Patra, P. K., Ciais, P., Arora, V. K., Bastviken, D.,

Bergamaschi, P., Blake, D. R., Brailsford, G., Bruhwiler, L., Carlson, K. M., Carrol, M., Castaldi, S., Chandra, N., Crevoisier, C., Crill, P. M., Covey, K., Curry, C. L., Etiope, G., Frankenberg, C., Gedney, N., Hegglin, M. I., Höglund-Isaksson, L., Hugelius, G., Ishizawa, M., Ito, A., Janssens-Maenhout, G., Jensen, K. M., Joos, F., Kleinen, T., Krummel, P. B., Langenfelds, R. L., Laruelle, G. G., Liu, L., Machida, T., Maksyutov, S., McDonald, K. C., McNorton, J., Miller, P. A., Melton,

J. R., Morino, I., Müller, J., Murguia-Flores, F., Naik, V., Niwa, Y., Noce, S., O'Doherty, S., Parker, R. J., Peng, C., Peng, S., Peters, G. P., Prigent, C., Prinn, R., Ramonet, M., Regnier, P., Riley, W. J., Rosentreter, J. A., Segers, A., Simpson, I. J., Shi, H., Smith, S. J., Steele, L. P., Thornton, B. F., Tian, H., Tohjima, Y., Tubiello, F. N., Tsuruta, A., Viovy, N., Voulgarakis, A., Weber, T. S., van Weele, M., van der Werf, G. R., Weiss, R. F., Worthy, D., Wunch, D., Yin, Y.,

Yoshida, Y., Zhang, W., Zhang, Z., Zhao, Y., Zheng, B., Zhu, Q., Zhu, Q., and Zhuang, Q.: The Global Methane Budget 2000–2017, Earth System Science Data, 12, 1561-1623, 10.5194/essd-12-1561-2020, 2020.

Sheng, J., Tunnicliffe, R., Ganesan, A. L., Maasakkers, J. D., Shen, L., Prinn, R. G., Song, S., Zhang, Y., Scarpelli, T., Anthony Bloom, A., Rigby, M., Manning, A. J., Parker, R. J., Boesch, H., Lan,

X., Zhang, B., Zhuang, M., and Lu, X.: Sustained methane emissions from China after 2012 despite declining coal production and rice-cultivated area, Environmental Research Letters, 16, 10.1088/1748-9326/ac24d1, 2021.

Yu, X., Millet, D. B., Wells, K. C., Henze, D. K., Cao, H., Griffis, T. J., Kort, E. A., Plant, G., Deventer, M. J., Kolka, R. K., Roman, D. T., Davis, K. J., Desai, A. R., Baier, B. C., McKain,

K., Czarnetzki, A. C., and Bloom, A. A.: Aircraft-based inversions quantify the importance of wetlands and livestock for Upper Midwest methane emissions, Atmos Chem Phys, 21, 951-971, 10.5194/acp-21-951-2021, 2021.

Wunch, D., Toon, G. C., Blavier, J. F., Washenfelder, R. A., Notholt, J., Connor, B. J., Griffith, D. W., Sherlock, V., and Wennberg, P. O.: The total carbon column observing network, Philos Trans

A Math Phys Eng Sci, 369, 2087-2112, 10.1098/rsta.2010.0240, 2011.